# Multi-Objective Design Optimization of Flexible Manufacturing Systems Using Design of Simulation Experiments: A Comparative Study

Abdessalem Jerbi [1] , Wafik Hachicha [2],* , Awad M. Aljuaid [2] , Neila Khabou Masmoudi [3] and Faouzi Masmoudi [4]

[1] OLID Laboratory, Higher Institute of Industrial Management of Sfax (ISGIS), University of Sfax, Sfax 3021, Tunisia; abdessalem.jerbi@isgis.usf.tn
[2] Department of Industrial Engineering, College of Engineering, Taif University, P.O. Box 11099, Taif 21944, Saudi Arabia; amjuaid@tu.edu.sa
[3] LASEM Laboratory, Department of Mechanical Engineering, National Engineering School of Sfax (ENIS), University of Sfax, Sfax 3038, Tunisia; neila.masmoudikhabou@enis.tn
[4] LA2MP Laboratory, Department of Mechanical Engineering, National Engineering School of Sfax (ENIS), University of Sfax, Sfax 3038, Tunisia; faouzi.masmoudi@enis.tn
* Correspondence: wafik.hachicha@isgis.usf.tn; Tel.: +966-53-194-0695

**Abstract:** One of the basic components of Industry 4.0 is the design of a flexible manufacturing system (FMS), which involves the choice of parameters to optimize its performance. Discrete event simulation (DES) models allow the user to understand the operation of dynamic and stochastic system performance and to support FMS diagnostics and design. In combination with DES models, optimization methods are often used to search for the optimal designs, which, above all, involve more than one objective function to be optimized simultaneously. These methods are called the multi-objective simulation–optimization (MOSO) method. Numerous MOSO methods have been developed in the literature, which spawned many proposed MOSO methods classifications. However, the performance of these methods is not guaranteed because there is an absence of comparative studies. Moreover, previous classifications have been focused on general MOSO methods and rarely related to the specific area of manufacturing design. For this reason, a new conceptual classification of MOSO used in FMS design is proposed. After that, four MOSO methods are selected, according to this classification, and compared through a detailed case study related to the FMS design problem. All of these methods studied are based on Design of Experiments (DoE). Two of them are metamodel-based approaches that integrate Goal Programming (GP) and Desirability Function (DF), respectively. The other two methods are not metamodel-based approaches, which integrate Gray Relational Analysis (GRA) and the VIKOR method, respectively. The comparative results show that the GP and VIKOR methods can result in better optimization than DF and GRA methods. Thus, the use of the simulation metamodel cannot prove its superiority in all situations.

**Keywords:** flexible manufacturing system; multi-objective simulation-optimization method; discrete event simulation; design of experiments; simulation metamodel; goal programming; desirability function; grey relations analysis; VIKOR method

## 1. Introduction

The fourth industrial revolution, known as industry 4.0, is considered the upcoming significant technology development as it allows customers to receive their products based on their expectations in terms of product varieties and quantiles [1]. Industry 4.0 can be attributed to its broadening focus on automation, decentralization, system integration, cyber-physical systems, etc. [2]. One of the basic components of Industry 4.0 is the Flexible Manufacturing System (FMS), which is an advanced production system that interconnects

machines, workstations, and logistics equipment, the entire manufacturing process being coordinated with the computer. FMS is intended for manufacturing tasks of large typological diversity, for high complexity, for ensuring timely delivery, and for minimal manufacturing costs, while production is unpredictable, organized in small batches, and has frequent changes [3].

Discrete Event Simulation (DES) is a powerful tool for analyzing and optimizing FMS for the purpose of design, modeling, and ongoing performance [4]. A simulation of an entire manufacturing system involves the identification of organization machines, robots, and the layout of the system, also involving multiple processes in the system. In particular, the design of FMS involves the choice of parameters and their value to optimize its performance. DES models allow the user to understand system performance and assist in behavior prediction and to support FMS diagnostics and design. However, DES responds to what-if questions as a tool for system evaluation; by itself, it cannot provide answers to how-if questions [5]. Moreover, DES is essentially a trial-and-error approach and is, therefore, time consuming and does not provide a method for optimization. In fact, many researchers have attempted to combine simulation and optimization procedures to provide a complete design solution with desired properties [6]. The problem of locating the most preferred alternative system design by using experimental evaluations performed using a computer DES is known as the Simulation Optimization (SO) problem.

The main classification criterion for SO approaches is the number of output performance measures. There are two groups of SO methods [7]. The first group is named Single-Objective Simulation Optimization (SOSO) approaches, which are focused on optimizing a single performance measure. The second group, which is studied in this research, covers Multi-Objective Simulation Optimization (MOSO) approaches. MOSO is an area of decision making of multiple criteria that is concerned with mathematical optimization problems that involve more than one objective function to be optimized simultaneously.

In the next section, the classifications of MOSO methods in the literature are presented, and then the MOSO methods used in FMS design are provided. After that, a new conceptual classification of MOSO was applied to FMS design. According to this classification, the aim is to justify the selection of four MOSO methods that are used in the comparative study.

### 1.1. Literature Overview

#### 1.1.1. General Literature Review of MOSO Methods

MOSO methods are an area of multiple-criterion decision making that optimize multiple performance measures via simulation. In MOSO literature, there are three main classification criteria for organizing these methods.

1. According to the articulation of the preferences of the Decision Maker (DM). This first classification criterion is proposed by Rosen et al. [8]. Four groups of methods are possible and include the following: (1) a priori MOSO methods when the DM expresses their preferences before optimization is conducted; (2) a posteriori MOSO methods (in these methods, the DM selects a solution at the end of the search. Although this approach avoids the disadvantage of the a priori approach by taking into account preference information only at the end of the optimization process, it can lead to extremely high computational costs); (3) a progressive articulation of DM preferences (also named Interactive MOSO Methods) (the progressive approaches repeatedly solicit preference information from the DM to guide the optimization process). These methods enable DM to change his preferences during the optimization process by incorporating knowledge that only becomes available during the search. Interactive methods may be useful when simulation runs are expensive and the DM is readily available to provide input. Finally, the fourth group involves (4) non preference MOSO methods that operate without regard to the preference of DM.

2. According to the research set and variables nature. This second classification criterion is proposed by Hunter et al. [7]. Three groups of methods are possible, including the following: (1) MOSO on finite sets, called Multi-Objective Ranking and Selection

(MORS); (2) MOSO with integer-ordered decision variables; and (3) MOSO with continuous decision variables. In the context of integer-ordered and continuous decision variables, we focus on methods that provably converge to a local efficient set under natural ordering. Furthermore, these methods of the three groups can also be viewed two groups according to the type of the final solution: global solution versus local solution [7]. The MORS methods provide a global solution, in which simulation replications are usually obtained from every point in the finite feasible set, and the estimated solution is the global estimated best. In addition, metaheuristics methods (named also random search) such as simulated annealing, Genetic Algorithms (GA), Tabu Search (TS), etc., also provide global solutions. Metaheuristics methods are efficient because they appropriately control stochastic error. However, the task is more challenging as it results in a number of solutions with different trade-offs among criteria, also known as Pareto optimal or efficient solutions.

3. According to the use or non-use of metamodels. This third classification is proposed implicitly in many research studies such as in Barton and Meckesheimer [9], do Amaral et al. [10], etc. A metamodel or model of the simulation model simplifies the SO in two ways: The metamodel response is deterministic rather than stochastic, and the run times are generally much shorter than the original simulation. The metamodel is used to identify and estimate the relationship between the inputs and outputs of the simulation model, forming a mathematical function that is used to evaluate possible solutions in the optimization process. For example, Hassannayebi et al. [11] highlight that the adoption of metamodel-based SO in industry and service problems has grown due to its potential to reduce the number of simulation rounds necessary in the optimization process. Note that the MOSO methods, which are based on the metamodel, also provide a global solution such as that discussed in the second classification criterion.

### 1.1.2. FMS Design Literature Review

The study of Diaz et al. [12] presents a MOSO approach for a reconfigurable production lines subject to scalable capacities. The production line produces two product families and is composed of 18 workstations. The authors utilized a Non-Dominated Sorting Genetic Algorithm II (NSGA-II), a variant of GA to address the assignment of the tasks to workstations and buffer allocation for simultaneously maximizing the Throughput Rate (TR) and minimizing total buffer capacity. Červeňanská et al. [13] explored an MOSO of an FMS via a scalar simulation-based optimization method. The authors integrated a simulation with Design of Experiment (DoE) and Weighted Sum and Product multi-objective methods to optimize the total number of products, the Mean Flow Time (MFT), the Machine UTILization (MUTIL), and the average costs per unit of part. The modeled FMS produces two different products with eight workstations using parallel automated work machines.

The paper of Hussain and Ali [14] studied the impact of four design and control factors, control architectures, sequencing flexibility, buffer capacity, and scheduling rule on the performance of an FMS. The studied FMS is composed of six Computer Numerical Control (CNC) machines producing six different types of parts. The system is evaluated on the basis of make-span, average MUTIL, and the average Waiting Time (WT) of parts at the queue using the Taguchi–Grey multi-objective method. Apornak et al. [15] considered a multi-objective optimization of five performance measures in FMS. The authors addressed the optimal set of queues capacity, queues discipline, conveyor and transporter's speed, and operational setup times in an FMS with objectives of minimization of the average WT of raw materials, two average Process Times (PT), as well as the transporter and assembler product outputs. The studied FMS is composed of three work stations producing various kinds of seats for the freight cars. Using DoE, the authors simulated and collected the performance measure of 36 random scenarios. Regression analysis was then used to

describe the metamodel of each performance measure. Consequently, the Response Surface Methodology (RSM) was applied to optimize the five objective functions.

Ahmadi et al. [16] proposed two Evolutionary Algorithms (EA): NSGA-II and NRGA are applied and compared to simultaneously combine the improvement of the make-span and stability of the schedule. This stability is evaluated by measuring the deviation of start and completion times of each job between prescheduled and realized schedule. The simulation is used to evaluate the state and condition of the machine breakdowns on a variety of manufacturing systems. Freitag and Hildebrandt [17] used a multi-objective simulation-based optimization to create a control strategy for an FMS by considering earliness and tardiness performance measures. This paper investigates the effect of 10 different attributes, which are the PT, the average PT of all waiting jobs, the Setup Time (ST), the average ST of all waiting jobs, the number of remaining operations, the time in system, the time in queue, the batch family size, the time until operational due date, and the average time until operational due date. The authors used the GA coupled with the simulation to solve the scheduling rule choice problem for a complex FMS.

Ammar et al. [18] investigated the size of the number of workers to be assigned to an FMS as well as the skills that each worker must have in a multi-objective optimization problem. The two objectives considered are minimizing the expected labor cost associated with the manufacturing team and minimizing the expected average task TR. The proposed multi-objective simulation optimization approach is applied to the design of teams of a manufacturing system; using the EA NSGA-II connected to a simulation model developed using Arena. Dengiz et al. [19] implemented a multi-objective optimization method of an FMS based on simulation through DoE, a regression meta-model, and the Goal Programming (GP) method. The authors have modeled and simulated by the ARENA simulation software an FMS with four workstations. Then, they applied the multi-objective optimization method to optimize the TR and MFT in the system by taking into consideration the number of operator, the velocity of material handling, the number of tool, scheduling rules, and the number of pallets as design and control parameters.

Using simulation results, Bouslah et al. [20] developed and solved a mathematical model based on RSM. The main objectives of the authors were to determine the optimal batch size, the optimal hedging level, and the economic sampling plan design, which minimized the average total holding cost, which includes the storage of the Work In Process (WIP) and final inventory stock, the average backlog cost, the average cost of sampling, the average costs of 100% inspection and rectification of the rejected batches, the average cost of transportation, and the average cost of replacement of non-conforming items sold to the consumer. However, the authors did not mention any details on the structure of the simulated manufacturing system. Iç et al. [21] considered a case study of simulation-based multi-objective optimization using the Technique for Order of Preference by Similarity to Ideal Solution (TOPSIS) method hybridized with the Taguchi design technique. The studied production system is an FMS department composed of four CNC machining centers and producing three part types. The authors based their optimization case on the cycle time, TR, and work in queue as performance measures. In addition, they used five factors as decision variables. These factors are the number of cutting tools, the number of operators, the number of pallets, the velocity of transporter robots, and the pallet selection strategy.

The paper of Wang et al. [22] applies an MOSO method to a flexible shop scheduling problem. The two investigated objective functions are the minimum of the maximum PT and the minimum of the maximum machine load. The main considered constraints are the production resources and the technological process. The scheduling model of an FMS is established using simulation software and integrated to NSGA-II EA. In Zhang et al. [23], a hybrid method based on hybrid GA and TS is used to address a multi-objective FMS scheduling problem. Two objectives, which are the make-span and the starting time deviations, are considered to improve schedule efficiency and stability. A case of study of six machines FMS was studied with four different job arrivals rate and six different number of job arrivals.

Azadeh et al. [24] integrated simulations with the GP method and DoE technique to address a multi-objective scheduling problem of an FMS. The proposed method was applied on a real textile shop floor to minimize make-span and tardiness. The authors determined the decision parameters by using the DoE technique by estimating the effects the dyeing machine type, the temperature of the printing, the temperature and the number of center machines, and the scheduling rules through meta-modeling. Then, they used GP to find the optimal values of these decision variables, which are subject to a set of technical and managerial constraints. Um et al. [25] presented the simulation based multi-objective optimization of the design of an FMS with Automated Guided Vehicles (AGVs). Their principal objectives were to minimize congestion and utilization and to maximize TR based on many parameters including the number, velocity, and dispatch rule of AGV, part types, scheduling, and buffer sizes. In this paper, the authors considered a nonlinear programming method combined to evolution strategy. Nonlinear programming was used to determine the design parameters of the system through multi-factorial and regression analyses, and an evolution strategy was used to verify each parameter for simulation-based optimization.

Syberfeldt et al. [26] describe the use of Artificial Neural Networks (ANN) and EA as MOSO methods to the manufacturing cell at Volvo Aero. The two investigated objectives were the maximization of cell utilization and the minimization of overdue components considering the component inter-arrival times and due date as decision criteria. Kuo et al. [27] proposed a practical case of the Grey-based Taguchi method as a MOSO method for a company that provides integrated circuit packaging services. The authors aimed to optimize TR and cycle time performance for ink marking machines to avoid backlog of orders or lost customers, and the TR of the system must be increased. They based their methodology on five three-level control factors, which are the PT, the machine buffer size, the time between adjustment, the ratio of the adjusted PT to original PT, and the mean time between failures.

Oyarbide-Zubillaga et al. [28] focused on the determination of the optimal preventive maintenance frequencies for multi-equipment systems. The authors apply simulation and NSGA-II to the multi-objective optimization problem of preventive maintenance activities to minimize the system's cost and to maximize profit by considering the production speed, the percentage of unavailability of a machine due to corrective maintenance, and the fraction of time before and after the last maintenance as control factors. The system cost was defined as the sum of the preventive and corrective maintenance, the production speed lost, and the quality costs for each of the machines. Profit is the result of selling non-defective products. Park et al. [29] presented a method for determining the design and control parameters of an FMS with multi-objective performance via a fully factorial DoE, regression analysis and trade-off programming. A hypothetical FMS with six workstations was modeled and simulated. The number, speed, and dispatching rules of AGVs, in addition to the number of pallets, the buffer sizes, and the loading, routing scheduling rules, were considered as control parameters. These eight parameters were simultaneously determined by compromising performance measures of TR, delay, MUTIL, and WIP that are formulated using regression analysis.

*1.2. The Proposed Conceptual Classification of MOSO for FMS Design*

There are many MOSO methods applied for FMS design. According to the previous literature review, it is better to classify them in three main groups: Group A, Group B, and Group C, as detailed in Table 1. This classification is applicable regardless of the articulation of DM's preferences. It should be noted that all of the previous MOSO methods that are applied in the design of FMS are global solutions.

**Table 1.** The proposed MOSO classification for FMS design.

| | The Use of DoE | The Use of Metamodel | Description |
|---|---|---|---|
| Group A | Yes | Yes | First, using designing simulation experiments. Second, applying optimization method on metamodel. A priori DM preferences are generally applied. |
| Group B | Yes | No | First, using designing simulation experiments. Second, applying multi criteria optimization method on experiments. A priori DM preferences are generally applied. |
| Group C | No | No | Iterative simulation and optimization using principally metaheuristics for random design research such as simulated annealing, genetic algorithms, etc. Only in this group, the articulation of the preferences of the DM is important. |

Table 2 summarizes the methods and techniques used in the MOSO methods used for FMS design. The presence of a cross "X" in a row and column intersection means that the research study stated in row use the method mentioned in column. It shows that all of the previous studies have applied a global solution method. These methods can be classified easily according the proposed classification in three groups (A, B, and C). Group C contains complex optimization techniques using metaheuristics, such as (GA, TS, EA, etc.). The performance of MOSO methods is not guaranteed because there is an absence of comparative studies. None of the previous studies has compared different MOSO methods.

**Table 2.** MOSO classification for FMS design (since year 2000).

| Study | Method | Number of Objectives | Number of Factors | A Priori | A Posteriori | Progressive | No-Preference | Ranking and Selection | Integer DV | Continuous DV | No Metamodel | Metamodel | Group A | Group B | Group C |
|---|---|---|---|---|---|---|---|---|---|---|---|---|---|---|---|
| [12] | NSGA II | 2 | 2 | | | | X | | | | X | | | | X |
| [13] | DoE, Weighted Sum, Weighted Product | 3 | 2 | X | | | | | X | X | | X | X | | |
| [14] | Taguchi design, GRA | 2 | 4 | | X | | | | X | | | | | X | |
| [15] | DoE, Regression metamodel, RSM | 5 | 8 | | | | X | | X | | | X | X | | |
| [16] | NSGA-II, NRGA | 2 | 2 | | X | | | | | | X | | | | X |
| [17] | Genetic Programming | 2 | 10 | | X | | | | | | X | | | | X |
| [18] | NSGA-II | 2 | 2 | | X | | | | | | X | | | | X |
| [19] | DoE, Regression meta-model, GP | 2 | 5 | X | | | | | | X | | X | X | | |
| [20] | RSM | 8 | 3 | | | | X | | | X | | X | X | | |
| [21] | Taguchi (DoE), TOPSIS | 3 | 5 | | | | X | X | | | X | | | X | |
| [22] | NSGA-II | 2 | 2 | | X | | | | | | X | | | | X |
| [23] | GA, TS | 2 | 2 | | | | X | | | | X | | | | X |

**Table 2.** *Cont.*

| Study | Method | Number of Objectives | Number of Factors | The Decision Maker's Preferences | | | | The Research Set and Variables Nature | | | The Use of Meta-Model or Not | | The Proposed Classification | | |
|---|---|---|---|---|---|---|---|---|---|---|---|---|---|---|---|
| | | | | A Priori | A Posteriori | Progressive | No-Preference | Ranking and Selection | Integer DV | Continuous DV | No Metamodel | Metamodel | Group A | Group B | Group C |
| [24] | DoE, Regression meta-model, GP | 2 | 5 | X | | | | | X | X | | X | X | | |
| [25] | Non-Linear Programming, Evolution Strategy | 3 | 6 | | | | X | | | | | X | | | X |
| [26] | EA, ANN | 2 | 2 | | X | | | | | | X | | | | X |
| [27] | Taguchi, GRA | 2 | 5 | | X | | | X | | | | X | | X | |
| [28] | NSGA-II | 2 | 3 | | X | | | | X | | X | | | | X |
| [29] | DoE, regression metamodel | 4 | 8 | X | | | | | | | | X | X | | |

### 1.3. The Objective of the Case Study

Our main contribution is to fill these gaps in the literature and to conduct a study of several relatively straightforward simulation-based FMS optimization methodologies that cover almost all categories of optimization methods classification. Our study investigates and compares the applicability and performances of the Goal Programming (GP), the Desirability Function (DF) method, the Grey Relational Analysis (GRA), and the VlseKriterijuska Optimizacija I Komoromisno Resenje (VIKOR) method.

All these methods are based on the DoE technique. They must be preceded by a design of experiments to program and sometimes analyze the simulation results. Moreover, these four multi-objective optimization methods have in common the type of preferences of DM; indeed, they are all based on an a priori decision of the DM for the choice of the objectives. On the other hand, the two methods GP and DF use the simulation-based metamodel technique and combine continuous and integer decision variables to solve the multi-objective optimization problem, while the two other methods, GRA and VIKOR, are based on the RS technique and use exclusively integer decision variables. The solutions reached by the GP and DF methods are then global and those reached by the GRA and VIKOR methods are local. In this study, we are interested in four multi-objective optimization methods in the context of FMS. An application on an FMS system will be used as a basis to compare the performances of these methods. It is mainly a matter of comparing the deviations between their results and the expected target values.

### 2. Materials and Methods

Figure 1 describes in detail the adopted MOSO methodologies applied to FMS. These methodologies are essentially made up of three stages. Each of these stages consists of various steps. In the first stage, the primary step starts with FMS factors levels and performance measures selection and definition. Consequently, DoE is constructed, and the corresponding simulation models are developed using ARENA 14 discrete event simulation software. In the final step of this first stage, simulations are run to collect data for every studied performance measure. These simulation results are then analyzed in the second phase by one of the four adopted multi-objective optimization methods. The steps of this phase are discussed in detail in the following paragraphs. Finally, the optimum factors levels are adopted in the last stage of the multi-objective optimization method.

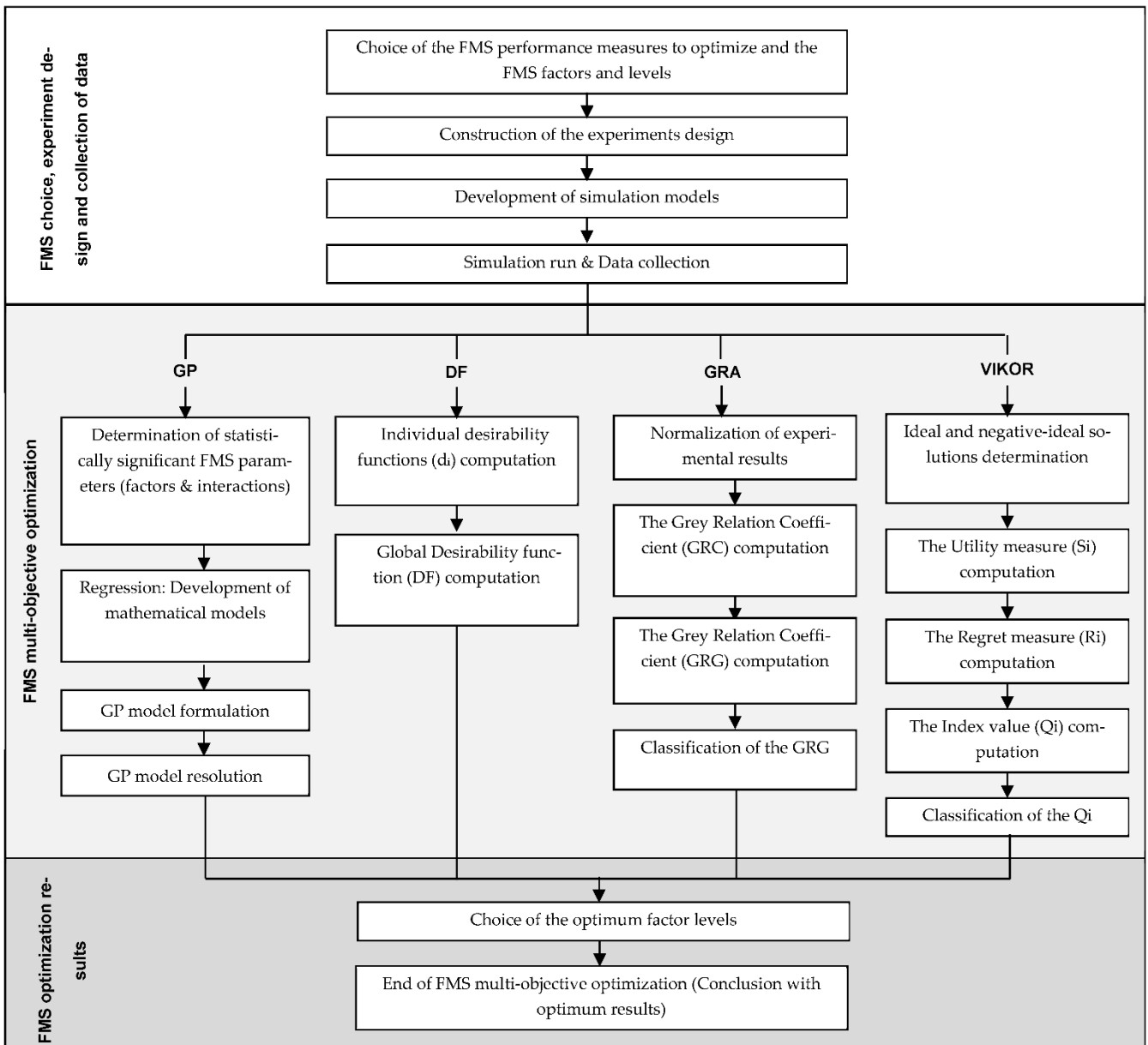

**Figure 1.** The major steps of the FMS optimization approach with multiple performance objectives.

### 2.1. Simulation of Possible FMS Designs according to the DoE

#### 2.1.1. The Case Study

The FMS investigated in this study is inspired by Pitchuka et al. [30]. An FMS is a manufacturing system characterized by a certain flexibility that allows the system to react in the case of changes. This flexibility is considered to fall into two categories. The first one, called routing flexibility, generally covers the system's ability to be changed to produce new product types. The second category is called machine flexibility, which consists of the ability to use various machines to perform the same manufacturing operation on a part.

- To capture the FMS flexibility effect on its performance, this research adopts different machine LAYOUT (LAYOUT) for the studied FMS. Indeed, Functional Layout (FL) and Cellular Layout (CL) are the two most used machine layouts in FMSs. In FL, functionally similar machines are grouped into departments, and all machines of every department can perform production operations for any incoming part [31]. However, CL is made up of independent manufacturing cells. Each of these cells is made up

of different machine types dedicated to the treatment of similar parts grouped into families. In addition, this work aimed also to measure the effect of part Batch Size (BS), part Inter-Arrival Time (IAT), and scheduling RULEs (RULE) on FMS performances (Table 3). The IAT, defined as the difference between the arrival times to the FMS of two consecutive parts, is generally generated by common probabilistic laws. In addition, parts are grouped into batches to reduce the machine's setup repetitions and transport times between work stations [31]. Furthermore, parts arriving at any work station are made to wait in a queue until the required machine becomes available. Once this required machine is idle, parts must be selected from the waiting queue based on scheduling rules [32–34]. As shown in Table 3, each of the considered FMS factors considered is studied with 2 levels.

**Table 3.** Studied factors and levels for the FMS case study.

| Factor | Levels | |
|---|---|---|
| | **1** | **2** |
| LAYOUT | FL | CL |
| IAT (Minutes/part) | 5 | 25 |
| BS (parts) | 5 | 10 |
| RULE | FCFS | SPT |

FL: functional layout; CL: cellular layout; IAT: inter-arrival time; BS: batch size; Rule: dispatching rule; FCFS: first come first served; SPT: short processing time.

- The FMS considered is composed of 8 machines grouped into 3 departments in FL and 2 cells in CL. The two departments "M" and "L" are composed of 3 machines each, while department "M" comprises only 2 machines. This MS is also characterized by two-part families composed of each of 2 part types. Each type of part requires 2 to 5 manufacturing operations (Table 4).

**Table 4.** Part routing used for the FMS case study.

| Part | | Functional Layout | Cellular Layout | |
|---|---|---|---|---|
| **Type** | **Family** | **Routing Departments** | **Cell** | **Routing Machines** |
| P1 | F2 | "L"→ "M"→"D" | C2 | "L2"→ "M2"→ "D2" |
| P2 | F1 | "L"→"D"→"M" | C1 | "L1"→ "D1"→ "M1" |
| P3 | F1 | "L"→"M" | C1 | "L1"→ "M1" |
| P4 | F2 | "L"→"M"→ "D"→ "L"→ "M" | C2 | "L2"→ "M2"→ "D2"→ "L3"→ "M3" |

- The setup and processing times for each type of part are provided in Table 5. Setup times on every machine can be reduced or cancelled by the setup factor ($\delta$) depending on the similarity of the successive parts family or type. Indeed, if successive parts belong to the same family, the subsequent part setup time must be reduced by a factor of $\delta = 0.5$. On the other hand, if these successive parts have the same type, no machine setup is needed and the subsequent part setup time must be cancelled by a factor of $\delta = 0$. Transfer times in the two layouts follow a statistical uniform law between 10 and 16 min.

**Table 5.** Parts' setup and processing time used for the FMS case study.

|  |  | L | M | D | L | M |
|---|---|---|---|---|---|---|
| P1 | ST | T (39, 44, 49) | U (49, 69) | N (65, 15) | | |
| | PT | T (15, 18, 21) | U (9, 13) | N (29, 5) | | |
| P2 | ST | T (90, 101, 110) | U (64, 84) | N (107, 35) | | |
| | PT | T (17, 21, 25) | U (20, 28) | N (14, 5) | | |
| P3 | ST | T (80, 84, 88) | U (72, 92) | | | |
| | PT | T (23, 28, 32) | U (14, 18) | | | |
| P4 | ST | T (62, 66, 70) | U (50, 58) | N (101, 10) | T (33, 38, 45) | U (78, 98) |
| | PT | T (18, 20, 22) | U (25, 33) | N (12, 5) | T (20, 23, 26) | U (15, 23) |

ST: setup time; PT: processing time; N: normal distribution; U: uniform distribution; T: triangular distribution; all the times are in minutes.

- To characterize the fluidity of parts flow in FMS, different optimization studies used WIP and MFT as major performance measures [31]. WIP has mainly been measured as the number of parts in the system, and MFT is simply obtained by averaging all durations between every part exit times and entry times in FMS. The TR of the production was adopted as the third performance measure. To evaluate TR, it is normal to measure the number of processed parts per unit of time. The maximization of such a measure of performance reflects the best use of material and human resources. To enhance the efficiency of FMS piloting, various optimization studies used the waiting and transfer times (WT and TT) as performance indicators, and they essentially aimed to minimize these two indicators.

### 2.1.2. The Simulation Model

FMS simulation models were built using Arena 14.0 software. The FL and CL models are composed of three parts: "Parts arriving", "Departments" or "Cells", and "System exit":

- Parts enter to the system through a "Create" module named "Parts Arrival" in which the BS and IAT times are specified. Then, they are grouped into batches by a "Batch" module, named "Arrival Parts Grouping", to assign them their corresponding types through an "Assign" module named "Part Type". Due to the stochastic nature of their PT and ST, these batches are separated into unit products through a "Separate" module called "Parts Separation" to assign them each of their execution times through one of the four "Assign" modules named "Attribute Part i". However, a preliminary step must be performed through a "Decide" module called "Parts Sorting" to direct each type of product to the corresponding "Assign" module. The products then proceed through a "Batch" module named "Parts Grouping" before proceeding through the "Route" module named "Transfer to System" (Figure 2).

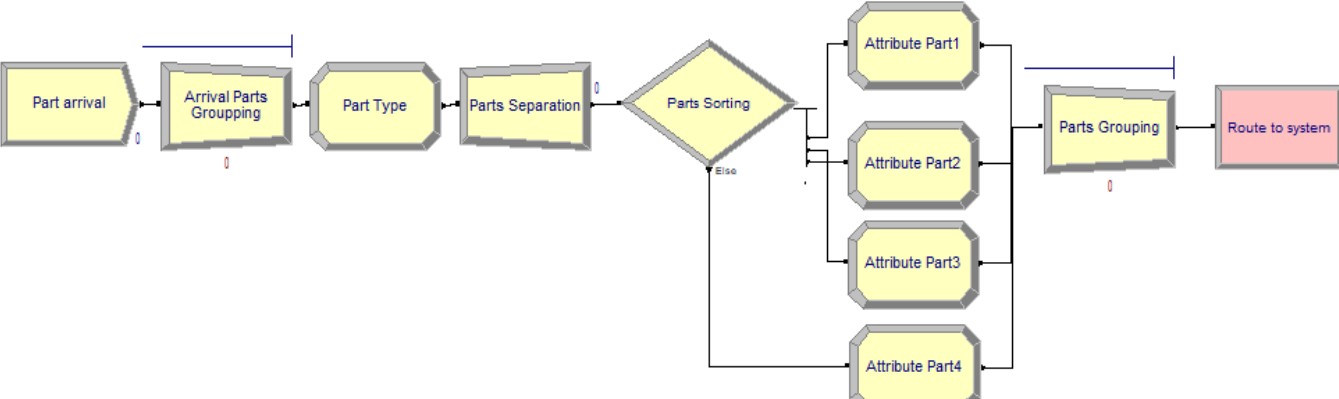

**Figure 2.** Simulation part model related to parts arriving.

- As soon as one products batch arrives in one of the departments, it is separated into unit products and put on hold in the department queue via the "Hold" module named "Waiting Queue Department i". This queue is governed by a "Queue" module in which the scheduling rule must be specified. Once one of the department machines becomes free, the selected waiting product is released from the "Hold" module. It then passes through a test, represented by the module "Decide" named "Machine Selection", which affects it toward this free machine. The processed products of the machines are grouped again in batches by the "Batch" module named "Grouping of Processed Parts Department i", which succeeds these machines. Finally, each batch of products is transferred to the next step in its production sequence through the module called "Route Department i" (Figure 3).

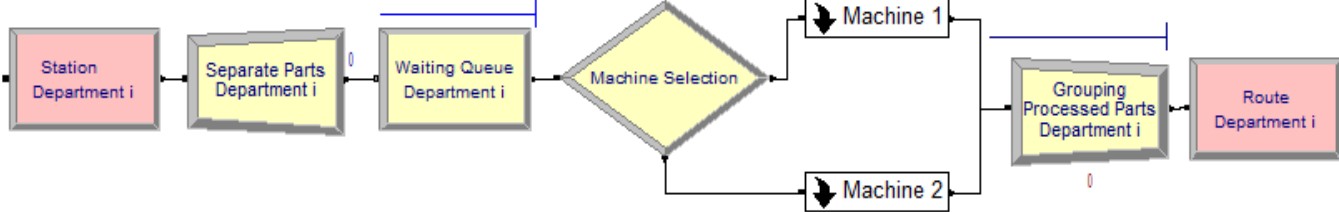

**Figure 3.** Simulation part model related to department "i".

In the case of the CL simulation model, as soon as a batch of products arrives in one of the cells, it is directed to the first machine in its production sequence. This batch is then separated into unit products by a "Separate" module called "Separation Parts Machine i". These products are then placed on hold in the queue of the machine via a "Hold" module named "Waiting Queue Machine i" until this machine becomes available. The choice of products from the machine queue is made according to the priority rule defined in the "Queue" module corresponding to this "Hold" module. The processed products by one of the machines are grouped into batches via the "Batch" module called "Grouping Parts Machine i". This batch is transferred to the next machine in its production sequence via the "Intracellular Route Cell i" module. By using this module, the transfer is performed in the cells, and the transfer time in this case is equal to zero. Each product with a completed production sequence must be evacuated to the system's output section. Hence, the "Cell i Output Route" module is used with a non-zero transfer time (Figure 4).

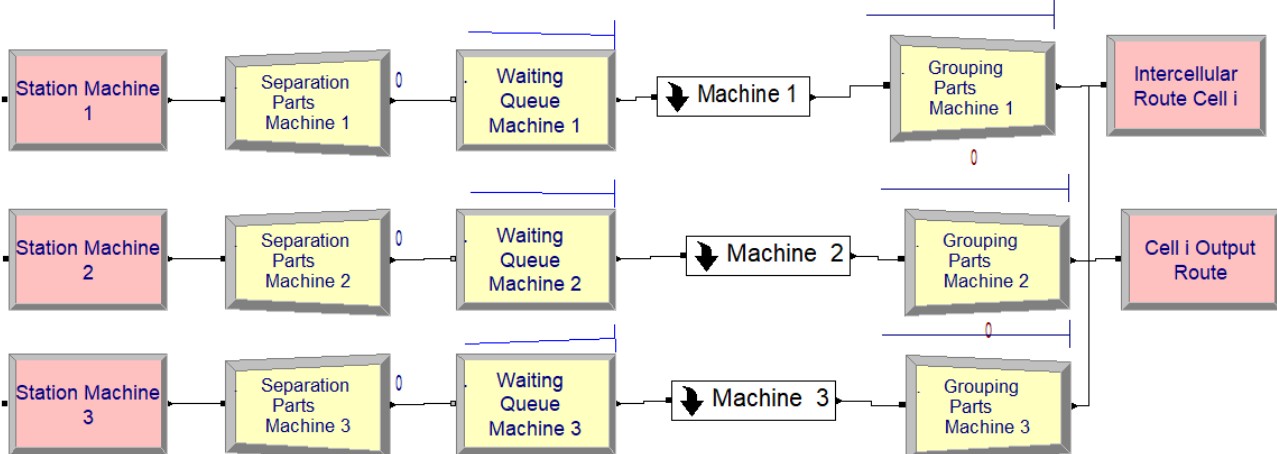

**Figure 4.** Simulation part model related to cell "i".

- In the two FL and CL simulation models, the machines are modeled by "Process" modules. In these modules, the transformation times are defined, which are function of PT and ST weighted by factor δ. Thus, Transformation time = PT + δxST. The value of factor δ depends on the similarity of the types of products entering and leaving the machine. In fact, a module called "Selection Delta Value Machine Selection i" applies a test on all incoming products to the machine to look for the value of this factor. For this, it compares two variables named "Part Type" and "Part Family" defined in the two "Assign" modules, named "Part Type in Machine i" and "Part Type Out Machine i". If the two variables "Part Type" are identical, the module "Delta value machine selection i" directs the incoming product to the module "Assign" named "Delta Equal 0 Machine i" corresponding to the value of factor δ = 0. If the two variables "Part Type" are different but the two variables "Part Family" are identical, the module "Delta Value Machine i" directs the incoming product to the module "Assign" named "Delta Equal 0.5 Machine i" corresponding to the value of factor δ = 0.5. Otherwise, module "Selection Delta Value Selection Machine i" directs the incoming product to the module "Assign" named "Delta Equal 1 Machine i", corresponding to the value of factor δ = 1 (Figure 5).

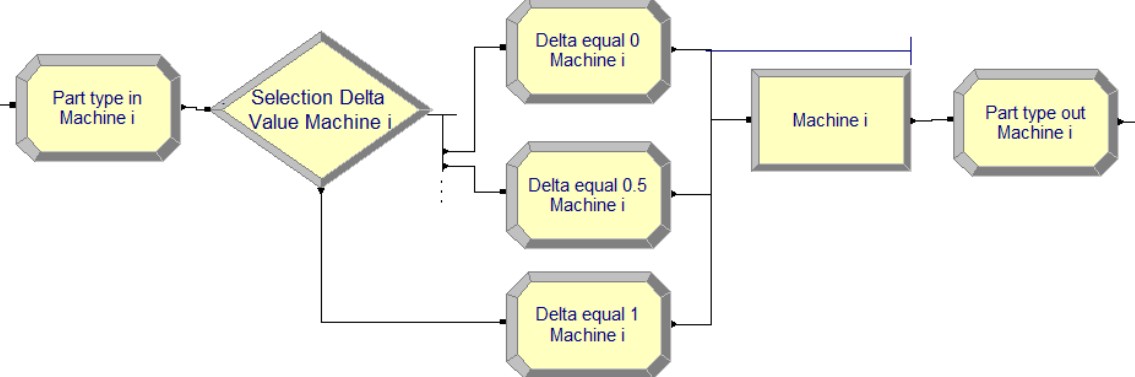

**Figure 5.** Simulation part model related to machine "i".

- The leaving products batch proceeds through an "Assig" module, called "Output Performance Measures", for computing and updating all variables defined as performance measures. The acquired data are then stored in an Excel file using a "Readwrite" module for further treatment and analysis. Finally, the batches of products are evacuated from the simulation model via the "Dispose" module named "System Exit" (Figure 6).

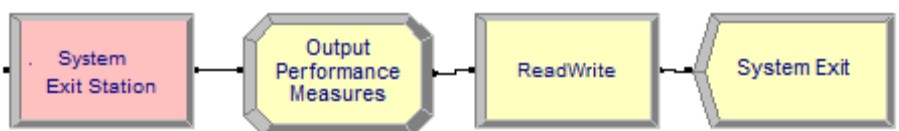

**Figure 6.** Simulation part model related to system exit.

## 2.2. Design of Experiments (DoE)

In this phase, we determine the number of distinct model settings to be run and the specific values of the factors for each of these simulation runs. There are many strategies for selecting the number of runs and the factor settings for each run include the following: random designs, combinatorial designs, sequential designs, factorial designs, etc.

Factorial designs are based on a grid, with each factor tested in combination with every level of every other factor. Factorial designs are attractive for three reasons: (i) The number of levels that are required for each factor is one greater than the highest-order power of that variable in the model, and the resulting design permits the estimation of

coefficients for all cross-product terms; (ii) they are probably the most commonly used class of designs; and (iii) the resulting set of run conditions are easy to visualize graphically for as many as nine factors [35].

The case study is about an FMS design with four factors, and each factor has two levels, as mentioned in the Table 3. Therefore, a $2^4$ full factorial design was used to collect simulation results.

### 2.3. Multi-Objective Optimization Methods

#### 2.3.1. The GP Method

The GP is an optimization technique to solve problems with variety of objectives, which are generally incommensurable and often conflict each other in a decision-making horizon. The standard version of GP was first introduced by Charnes and Coper [31]. The GP model is based on an objective function formulated to find the most satisfactory solution that minimizes the total sum of positive and negative deviations from the level of attainment of the objectives levels (goals) set by the decision maker. This objective function is subject to physical and operating constraints of the system. The first type of constraints represents operating physical limits of the studied system. As for the second constraints, they are generally described by mathematical connections between the FMS factors and interactions and the performance measures to optimize. Hence, the principal purpose of the second stage of two first steps of the DoE-GP hybridization method is to build mathematical connections between FMS factors and responses. Statistical analyses are applied on the obtained simulation results to identify significant factors and interactions, and the relationships between the identified significant factors and interactions and the performance measures are translated to mathematical models by using the regression technique. In the third step of this stage, the GP model is developed setting the performance measures as goals and including other FMS constraints. Finally, this model is resolved using resolved using LINGO 18.0 software. The aim of this GP model is to find the most suitable levels of FMS factors that lower the total deviation of each performance measure from their respective target levels obtained in DoE.

The GP model takes the following form.

$$\text{Minimise } Z = \sum_{i=1}^{p} \delta_i^+ + \delta_i^-, \tag{1}$$

Moreover, it is subject to the following:

$$\sum_{j=1}^{n} a_{ij}x_j - \delta_i^+ + \delta_i^- = g_i \ (i = 1\ldots p), \tag{2}$$

$$\rho x \leq C \ (\text{the operating physical constrain of the system}) \tag{3}$$

$$x_j \geq 0 \ (j = 1\ldots n), \tag{4}$$

$$\delta_i^+ \text{and} \delta_i^- \geq 0 \ (i = 1\ldots p), \tag{5}$$

where the following is the case:

1. $g_i$: The goal set for the ith objective for (i =1 ... p) (the objectives here are the performance measures);
2. $x_j$: The jth decision variable for (j = 1 ... n) (the decision variables here are the significant FMS factors and interactions);
3. $a_{ij}$: The technological parameters (these parameters are the coefficients of the developed mathematical models relating the performance measures to significant FMS factors and interactions);
4. $\rho$: The matrix of coefficients related to the physical FMS constraints;
5. C: The vector of available physical FMS resources;
6. $\delta_i^+$, $\delta_i^-$ : The positive and negative deviations from the goals values.

### 2.3.2. The DF Method

The DF method is based on two steps. The first defines a desirability function by assigning values to responses that reflect their desirability. This involves transforming each value of the performance measure 'j' of experiment 'i', $y_{ij}$, into a partial dimensionless desirability function $d_i$, where $0 \leq d_i \leq 1$. This function includes the choices of the decision maker when constructing the optimization procedure.

A one-sided desirability transformation arises when the goal is to maximize or minimize the response, and two values A and B must be specified as the lower and upper limits. Equations (6) and (7) present the one-sided transformation equations that will be used for minimization and maximization goals, respectively.

$$d_i = \begin{cases} 1 & A \geq y_{ij} \\ \left( \frac{y_{ij}-A}{B-A} \right)^{\omega_j} & A \leq y_{ij} \leq B \\ 0 & y_{ij} \geq B \end{cases}, \tag{6}$$

$$d_i = \begin{cases} 1 & y_{ij} \geq B \\ \left( \frac{y_{ij}-A}{B-A} \right)^{\omega_j} & A \leq y_{ij} \leq B \\ 0 & A \geq y_{ij} \end{cases}, \tag{7}$$

The parameter $\omega_j$ can be described as a power value or weight allocated according to the researcher subjective impression about the role of the response in the total desirability of the product.

A value of $\omega_j$ equal to 1 implies that a linear desirability function is applied. If the value of $\omega_j$ is less than 1, the obtained desirability function means that performance does not have to be close to the lower or upper limit, depending on the optimization goal, to have a higher desirability value. In contrast, if the value $\omega_j$ is greater than 1, the desirability function implying that the performance has to be closest to the lower or upper limit, depending on the optimization goal, to have a higher desirability value.

To simultaneously optimize multiple performance sets, the individual desirability is combined using a geometric mean in the composite desirability.

$$DF = \left( \prod_{i=1}^{n} d_i \right)^{\frac{1}{n}}, \tag{8}$$

A value of DF different from zero implies that all performances are in a desirable range simultaneously. In addition, a value of DF close to 1 means that the combination of the different criteria is globally optimal and the performances values are near the target values.

### 2.3.3. The GRA Method

Units of performance measurement are often different, so the influence of some of them may be neglected. This can also happen if some performance measures have a very wide range compared to others. In addition, if the expected optimization goals are contradictory, this will result in incorrect results in the analysis [36]. It is, therefore, necessary to normalize all performance values for each experiment in the first step of the multi-objective GRA-based optimization method's second stage.

In the developed DoE, for each of the "m" simulation experiments, "n" performance measures are measured. The ith experiment trial can be expressed as $Y_i = (y_{i1}, y_{i2}, \ldots, y_{ij}, \ldots, y_{in})$. Here, $y_{ij}$ is the value of the performance measure "j" of experiment "i". The term $Y_i$ can be translated into the comparability sequence $X_i = (x_{i1}, x_{i2}, \ldots, x_{ij}, \ldots, x_{in})$

using one of Equations (9) and (10), which are, respectively, used for larger-the-better and smaller-the-better objective values:

$$x_{ij} = \frac{y_{ij} - \underline{y_j}}{\overline{y_j} - \underline{y_j}} i = 1, 2, \ldots, m j = 1, 2, \ldots, n, \tag{9}$$

$$x_{ij} = \frac{\overline{y_j} - y_{ij}}{\overline{y_j} - \underline{y_j}} i = 1, 2, \ldots, m j = 1, 2, \ldots, n, \tag{10}$$

where the following is the case.

$$\overline{y_j} = \text{Max}\left\{y_{ij}, i = 1, 2, \ldots, m\right\}, \tag{11}$$

$$\underline{y_j} = \text{Min}\left\{y_{ij}, i = 1, 2, \ldots, m\right\}, \tag{12}$$

After the normalization procedure, all $x_{ij}$ values, relative to the performance measures, will be scaled in [0, 1]. The Grey Relational Coefficient (GRC) is then computed to determine how close $x_{ij}$ is to $x_{0j} = \text{Max}\{x_{ij}, i = 1, 2, \ldots, m\}$. The larger the grey relational coefficient, the closer $x_{ij}$ and $x_{0j}$ are. The grey relational coefficient can be calculated in the second step by the following:

$$\gamma\left(x_{0j}, x_{ij}\right) = \frac{\Delta_{min} + \zeta\Delta_{max}}{\Delta_{ij} + \zeta\Delta_{max}} i = 1, 2, \ldots, m j = 1, 2, \ldots, n, \tag{13}$$

where the following is the case.

$$\Delta_{ij} = \left|x_{0j} - x_{ij}\right|, \tag{14}$$

$$\Delta_{min} = \text{Min}\left\{\Delta_{ij}, i = 1, 2, \ldots, m; j = 1, 2, \ldots, n\right\}, \tag{15}$$

$$\Delta_{max} = \text{Max}\left\{\Delta_{ij}, i = 1, 2, \ldots, m; j = 1, 2, \ldots, n\right\}, \tag{16}$$

Note that $\zeta$ is the distinguishing coefficient, $\zeta \in [0, 1]$. The purpose of this coefficient is to expand or compress the range of the grey relational coefficient; usually, it is set equal to 0.5.

Once the entire GRC is computed, the Grey Relational Grade (GRG) is calculated in the third step based on the comparability and the reference sequence $X_i = (x_{i1}, x_{i2}, \ldots, x_{ij}, \ldots, x_{in})$ and $X_0 = (x_{01}, x_{02}, \ldots, x_{0j}, \ldots, x_{0n})$ using the following:

$$\Gamma(X_0, X_i) = \sum_{j=1}^{n} \omega_j \gamma\left(x_{0j}, x_{ij}\right); i = 1, 2, \ldots, m, \tag{17}$$

where $\omega_j$ is the weight for the jth response, chosen by the decision makers. Of course, the sum of $\omega_j$ is equal to 1.

In the final step of the GRA method, the GRG values are ranked in decreasing order. The optimal trial corresponds to the GRG maximum value.

2.3.4. The VIKOR Method

As in the case of the GRA method, which is based on GRG ranking, the VIKOR method is based on the computation of the VIKOR index and its ranking. In the first step of the VIKOR method, the ideal solution (A*) and the negative-ideal solution (A⁻) are to be determinate. A* and A⁻ represent, respectively, the maximum and minimum performance measure values of every experimental trial, and they are described as follows.

$$A^* = \text{Max}\left\{y_{ij}, i = 1, 2, \ldots, m\right\} = \left\{y_1^*, y_2^*, \ldots, y_j^*, \ldots, y_n^*\right\}, \tag{18}$$

$$A^- = \text{Min}\left\{y_{ij}, i = 1, 2, \ldots, m\right\} = \left\{y_1^-, y_2^-, \ldots, y_j^-, \ldots, y_n^-\right\}, \tag{19}$$

In the two following steps of the VIKOR method application the utility and the regret measures for the ith experimental trial, $S_i$ and $R_i$ respectively, are computed as follows:

$$S_i = \sum_{j=1}^{n} \omega_j \left( y_j^* - y_{ij} \right) / \left( y_j^* - y_j^- \right), \tag{20}$$

$$R_i = \text{Max}_i \left[ \omega_j \left( y_j^* - y_{ij} \right) / \left( y_j^* - y_j^- \right) \right], \tag{21}$$

where $\omega_j$ is the weight for the jth response, chosen by the decision makers. Of course, the sum of $\omega_j$ is equal to 1.

In the fourth step, the VIKOR index of the ith experimental trial is computed as follows:

$$Q_i = v \left[ \frac{S_i - S^-}{S^* - S^-} \right] + (1 - v) \left[ \frac{R_i - R^-}{R^* - R^-} \right], \tag{22}$$

where the following is the case.

$$S^- = \text{Min} S_i, \tag{23}$$

$$S^* = \text{Max} S_i, \tag{24}$$

$$R^- = \text{Min} R_i, \tag{25}$$

$$R^* = \text{Max} R_i, \tag{26}$$

Note that $v$ is the weight of the maximum group's utility. It is usually set to 0.5.

In the final step of the VIKOR method application, the VIKOR index values are ranked in decreasing order, and the optimal trials correspond to the maximum value.

## 3. Results

### 3.1. Simulation Results

The case study is about an FMS design with four factors, each factor has two levels, as mentioned in Table 3. Therefore, a $2^4$ full factorial design was used to collect simulation results. Each of the 16 simulation experiments was replicated 10 times. Simulation results show that a warm-up period of 10,000 min is needed, and models can then be run for 90,000 min. All final simulation results are provided in Appendix A. MFT simulation results are stated in Table A1, WIP simulation results are in Table A2, TR simulation results are in Table A3, WT simulation results are in Table A4, and TT simulation results are in Table A5.

### 3.2. Multi-Objective Optimization Methods

#### 3.2.1. The GP Method

The use of GP as an MOSO method contains mainly four phases. The first phase is about the selection of the significant coefficient of the metamodel using Student's *t*-test. The second phase provides the final metamodel of each performance measure. The third and fourth phases concern the application of GP optimization.

1. Determination of statistically significant FMS parameters: The main effects of the studied factors and interactions were analyzed in $\alpha = 0.05$ of significance levels using the MINITAB statistical package (Table 6). Significant factors and interactions ($p \leq 0.05$) are shown in bold.

**Table 6.** Estimated coefficients of the simulation metamodel.

| Term | MFT | | | WIP | | | TR | | | WT | | | TT | | |
|---|---|---|---|---|---|---|---|---|---|---|---|---|---|---|---|
| | Coef | T | P | Coef | T | P | Coef | T | P | Coef | T | P | Coef | T | P |
| Constant | 2034.8 | 80.78 | **0.000** | 51.541 | 85.46 | **0.000** | 34.7833 | 647.60 | **0.000** | 12,804 | 51.20 | **0.000** | 243.865 | 1955.65 | **0.000** |
| BS | −883.4 | −35.07 | **0.000** | −32.921 | −54.59 | **0.000** | −8.6261 | −160.60 | **0.000** | −2499 | −9.99 | **0.000** | 81.199 | 651.16 | **0.000** |
| IAT | −1452.7 | −57.67 | **0.000** | −45.593 | −75.60 | **0.000** | −13.4205 | −249.86 | **0.000** | −9222 | −36.87 | **0.000** | −0.175 | −1.40 | 0.163 |
| RULE | −977.3 | −38.80 | **0.000** | 2.231 | 3.70 | **0.000** | 0.1465 | 2.73 | **0.007** | −4889 | −19.55 | **0.000** | 0.243 | 1.95 | 0.054 |
| LAYOUT | −1237.8 | −49.14 | **0.000** | −39.896 | −66.15 | **0.000** | 5.1784 | 96.41 | **0.000** | −7200 | −28.79 | **0.000** | −48.646 | −390.11 | **0.000** |
| BS*IAT | 967.3 | 38.40 | **0.000** | 33.700 | 55.88 | **0.000** | 1.5064 | 28.05 | **0.000** | 4323 | 17.29 | **0.000** | 0.015 | 0.12 | 0.903 |
| BS*RULE | 828.1 | 32.87 | **0.000** | −3.119 | −5.17 | **0.000** | −0.0592 | −1.10 | 0.273 | 3444 | 13.77 | **0.000** | 0.065 | 0.52 | 0.603 |
| BS*LAYOUT | 1032.5 | 40.99 | **0.000** | 34.680 | 57.50 | **0.000** | −4.7818 | −89.03 | **0.000** | 5362 | 21.44 | **0.000** | −15.899 | −127.50 | **0.000** |
| IAT*RULE | 977.6 | 38.81 | **0.000** | −2.155 | −3.57 | **0.000** | −0.1495 | −2.78 | **0.006** | 4887 | 19.54 | **0.000** | −0.111 | −0.89 | 0.376 |
| IAT*LAYOUT | 1258.3 | 49.95 | **0.000** | 40.047 | 66.40 | **0.000** | −5.1661 | −96.18 | **0.000** | 7831 | 31.31 | **0.000** | 0.026 | 0.21 | 0.836 |
| RULE*LAYOUT | 954 | 37.87 | **0.000** | −2.313 | −3.83 | **0.000** | −0.1110 | −2.07 | **0.041** | 4762 | 19.04 | **0.000** | −0.191 | −1.53 | 0.128 |
| BS*IAT*RULE | −829 | −32.91 | **0.000** | 3.065 | 5.08 | **0.000** | 0.0650 | 1.21 | 0.228 | −3447 | −13.78 | **0.000** | 0.024 | 0.19 | 0.850 |
| BS*IAT*LAYOUT | −941.1 | −37.36 | **0.000** | −33.724 | −55.92 | **0.000** | 4.7214 | 87.90 | **0.000** | −4624 | −18.49 | **0.000** | −0.066 | −0.53 | 0.600 |
| BS*RULE*LAYOUT | −808.6 | −32.10 | **0.000** | 3.316 | 5.50 | **0.000** | 0.0375 | 0.70 | 0.486 | −3366 | −13.46 | **0.000** | 0.070 | 0.56 | 0.577 |
| IAT*RULE*LAYOUT | −954.0 | −37.87 | **0.000** | 2.269 | 3.76 | **0.000** | 0.1120 | 2.08 | **0.039** | −4766 | −19.06 | **0.000** | 0.027 | 0.21 | 0.831 |
| BS*IAT*RULE*LAYOUT | 809.1 | 32.12 | **0.000** | −3.273 | −5.43 | **0.000** | −0.0413 | −0.77 | 0.444 | 3359 | 13.43 | **0.000** | −0.126 | −1.01 | 0.314 |

A*B: interaction between factor A and factor B (For example, BS*IAT means interaction between factor BS and factor IAT.

2.  Development of mathematical models: Based on the simulation results, mathematical models were computed. Regression Equations (27)–(31) with identified significant factors been derived for MFT, WIP, TT, and WT.

$$\begin{aligned}
\text{MFT} = {} & 144.633 - 13633 \times \text{BS} - 5742.6 \times \text{IAT} - 68028 \times \text{RULE} - 71627 \times \text{LAYOUT} \\
& + 542.4 \times \text{BS} \times \text{IAT} + 6511 \times \text{BS} \times \text{RULE} + 6809 \times \text{BS} \times \text{LAYOUT} + 2721.8 \times \text{IAT} \times \text{RULE} \\
& + 2845.2 \times \text{IAT} \times \text{LAYOUT} + 33807 \times \text{RULE} \times \text{LAYOUT} - 260.52 \times \text{BS} \times \text{IAT} \times \text{RULE} \\
& - 269.48 \times \text{BS} \times \text{IAT} \times \text{LAYOUT} - 3235.7 \times \text{BS} \times \text{RULE} \times \text{LAYOUT} - 1352.5 \times \text{IAT} \times \\
& \text{RULE} \times \text{LAYOUT} + 129.46 \times \text{BS} \times \text{IAT} \times \text{RULE} \times \text{LAYOUT}
\end{aligned} \tag{27}$$

*With $R^2$ = 99.41% and $R^2(adj)$ = 99.34%,*

$$\begin{aligned}
\text{WIP} = {} & 1078.3 - 96.84 \times \text{BS} - 42.75 \times \text{IAT} + 239.6 \times \text{RULE} - 529.2 \times \text{LAYOUT} + 3.849 \times \\
& \text{BS} \times \text{IAT} - 25.91 \times \text{BS} \times \text{RULE} + 48.47 \times \text{BS} \times \text{LAYOUT} - 9.52 \times \text{IAT} \times \text{RULE} + 20.99 \\
& \times \text{IAT} \times \text{LAYOUT} - 121.6 \times \text{RULE} \times \text{LAYOUT} + 1.031 \times \text{BS} \times \text{IAT} \times \text{RULE} - 1.912 \times \\
& \text{BS} \times \text{IAT} \times \text{LAYOUT} + 13.16 \times \text{BS} \times \text{RULE} \times \text{LAYOUT} + 4.835 \times \text{IAT} \times \text{RULE} \times \\
& \text{LAYOUT} - 0.5237 \times \text{BS} \times \text{IAT} \times \text{RULE} \times \text{LAYOUT}
\end{aligned} \tag{28}$$

*With $R^2$ = 99.47% and $R^2(adj)$ = 99.42%,*

$$\begin{aligned}
\text{TP} = {} & -61.61 + 9.883 \times \text{BS} + 4.1508 \times \text{IAT} + 2.415 \times \text{RULE} + 98.713 \times \text{LAYOUT} - 0.50632 \\
& \times \text{BS} \times \text{IAT} - 9.4912 \times \text{BS} \times \text{LAYOUT} - 0.0971 \times \text{IAT} \times \text{RULE} - 3.9333 \times \text{IAT} \times \\
& \text{LAYOUT} - 1.116 \times \text{RULE} \times \text{LAYOUT} + 0.37771 \times \text{BS} \times \text{IAT} \times \text{LAYOUT} + 0.0448 \times \text{IAT} \\
& \times \text{RULE} \times \text{LAYOUT}
\end{aligned} \tag{29}$$

*With $R^2$ = 99.88% and $R^2(adj)$ = 99.87%,*

$$\text{TT} = 3.12 + 51.558 \times \text{BS} - 1.898 \times \text{LAYOUT} - 12.7192 \times \text{BS} \times \text{LAYOUT} \tag{30}$$

*With $R^2$ = 99.97% and $R^2(adj)$ = 99.97%,*

$$\begin{aligned}
\text{WT} = {} & 669{,}420 - 58944 \times \text{BS} - 26657 \times \text{IAT} - 298872 \times \text{RULE} - 334431 \times \text{LAYOUT} + \\
& 2351 \times \text{BS} \times \text{IAT} + 27062 \times \text{BS} \times \text{RULE} + 30009 \times \text{BS} \times \text{LAYOUT} + 11951 \times \text{IAT} \times \text{RULE} \\
& + 13247 \times \text{IAT} \times \text{LAYOUT} + 148498 \times \text{RULE} \times \text{LAYOUT} - 1081.9 \times \text{BS} \times \text{IAT} \times \text{RULE} \\
& - 1176.1 \times \text{BS} \times \text{IAT} \times \text{LAYOUT} - 13447 \times \text{BS} \times \text{RULE} \times \text{LAYOUT} - 5937 \times \text{IAT} \times \\
& \text{RULE} \times \text{LAYOUT} + 537.4 \times \text{BS} \times \text{IAT} \times \text{RULE} \times \text{LAYOUT}
\end{aligned} \tag{31}$$

*With $R^2$ = 97.86% and $R^2(adj)$ = 97.64%,*

Every constant in each of these equations corresponds to the average responses for each performance measure, and the coefficients assigned to the factors and interactions correspond to their respective effects.

3.  GP model formulation and resolution: We propose a GP model in which the selected performance measures are considered. The optimal configuration of decision variables minimizes the sum of penalties (dj). The parameter dj are deviations from the desired levels of the goals that are subject to series constraints. With the regression equations presented previously, the above-mentioned goal programming model can be stated as shown in Equations (32)–(40):

$$\text{Min Z} = d^+_{\text{MFT}} + d^+_{\text{WIP}} + d^-_{\text{TR}} + d^+_{\text{WT}} + d^+_{\text{TT}}, \tag{32}$$

which are subject to the following.

$$144633 - 13633 \times BS - 5742.6 \times IAT - 68028 \times RULE - 71627 \times LAYOUT + 542.4 \times BS \times$$
$$IAT + 6511 \times BS \times RULE + 6809 \times BS \times LAYOUT + 2721.8 \times IAT \times RULE + 2845.2 \times$$
$$IAT \times LAYOUT + 33807 \times RULE \times LAYOUT - 260.52 \times BS \times IAT \times RULE - 269.48 \times \quad (33)$$
$$BS \times IAT \times LAYOUT - 3235.7 \times BS \times RULE \times LAYOUT - 1352.5 \times IAT \times RULE \times$$
$$LAYOUT + 129.46 \times BS \times IAT \times RULE \times LAYOUT + d_{MFT}^{-} - d_{MFT}^{+} = G_{MFT},$$

$$1078.3 - 96.84 \times BS - 42.75 \times IAT + 239.6 \times RULE - 529.2 \times LAYOUT + 3.849 \times BS \times$$
$$IAT - 25.91 \times BS \times RULE + 48.47 \times BS \times LAYOUT - 9.52 \times IAT \times RULE + 20.99 \times IAT \times$$
$$LAYOUT - 121.6 \times RULE \times LAYOUT + 1.031 \times BS \times IAT \times RULE - 1.912 \times BS \times IAT \times \quad (34)$$
$$LAYOUT + 13.16 \times BS \times RULE \times LAYOUT + 4.835 \times IAT \times RULE \times LAYOUT - 0.5237 \times$$
$$BS \times IAT \times RULE \times LAYOUT + d_{WIP}^{-} - d_{WIP}^{+} = G_{WIP},$$

$$-61.61 + 9.883 \times BS + 4.1508 \times IAT + 2.415 \times RULE + 98.713 \times LAYOUT - 0.50632 \times BS \times$$
$$IAT - 9.4912 \times BS \times LAYOUT - 0.0971 \times IAT \times RULE - 3.9333 \times IAT \times LAYOUT - \quad (35)$$
$$1.116 \times RULE \times LAYOUT + 0.37771 \times BS \times IAT \times LAYOUT + 0.0448 \times IAT \times RULE \times$$
$$LAYOUT + d_{TR}^{-} - d_{TR}^{+} = G_{TR},$$

$$3.12 + 51.558 \times BS - 1.898 \times LAYOUT - 12.7192 \times BS \times LAYOUT + d_{TT}^{-} - d_{TT}^{+} = G_{TT}, \quad (36)$$

$$669420 - 58944 \times BS - 26657 \times IAT - 298872 \times RULE - 334431 \times LAYOUT + 2351 \times BS \times$$
$$IAT + 27062 \times BS \times RULE + 30009 \times BS \times LAYOUT + 11951 \times IAT \times RULE + 13247 \times IAT \times \quad (37)$$
$$LAYOUT + 148498 \times RULE \times LAYOUT - 1081.9 \times BS \times IAT \times RULE - 1176.1 \times BS \times IAT \times$$
$$LAYOUT - 13447 \times BS \times RULE \times LAYOUT - 5937 \times IAT \times RULE \times LAYOUT + 537.4 \times BS \times$$
$$IAT \times RULE \times LAYOUT + d_{WT}^{-} - d_{WT}^{+} = G_{WT},$$

$$LAYOUT \text{ and } RULE \text{ are binary (1 or 2)}, \quad (38)$$

$$5 \le IAT \le 25, \quad (39)$$

$$5 \le BS \le 10, \quad (40)$$

The objective $G_{MFT}$, $G_{WIP}$, $G_{TR}$, $G_{TT}$, and $G_{WT}$ goal values were fixed basing on the experimental design results.

4.  The GP model was solved using the mathematical software LINGO 18.0. The best value of the objective function was found to be equal to 136.99 and was obtained for the following levels of the studied factors: LAYOUT = CL, RULE = FCFS, IAT = 25, and BS = 5.

### 3.2.2. The DF Method

Applying Equations (6) and (7) for the studied performance measures, the individual desirability functions 'd' are very close to 1.0, as shown in Figure 7. Furthermore, Figure 7 illustrates the effect of each factor (columns) on the FMSs' performance measures and the desirability of the composite (rows). The red vertical lines and the corresponding numbers in red indicate the levels of optimal factors. The blue horizontal lines and the corresponding numbers in blue represent the values of the performance measures corresponding to the levels of optimal factors. Each of the performance measures is accompanied by the corresponding desirability function values '$d_i$'. In addition, the first row provides the value of the composite desirability 'DF', as presented in Equation (8), corresponding to the levels of the optimal factors. The obtained DF is equal to 0.984, which represents an ideal case of optimization. To obtain this desirability, the factors' levels must be set to the values shown below the global solution in Figure 7. That is, BS = 5, IAT = 5, LAYOUT = CL, and RULE = SPT.

### 3.2.3. The GRA Method

Based on Equations (9)–(17), the simulation results were normalized, and the GRC and GRG were calculated (Table 7). Once GRG was ranked, it appears that the optimum performance measures were obtained for the factor levels LAYOUT = CL, RULE = SPT, IAT = 5, and BS = 5. The row in bold in Table 7 indicate the optimal solution obtained using the GRA method, which has a rank equal to 1.

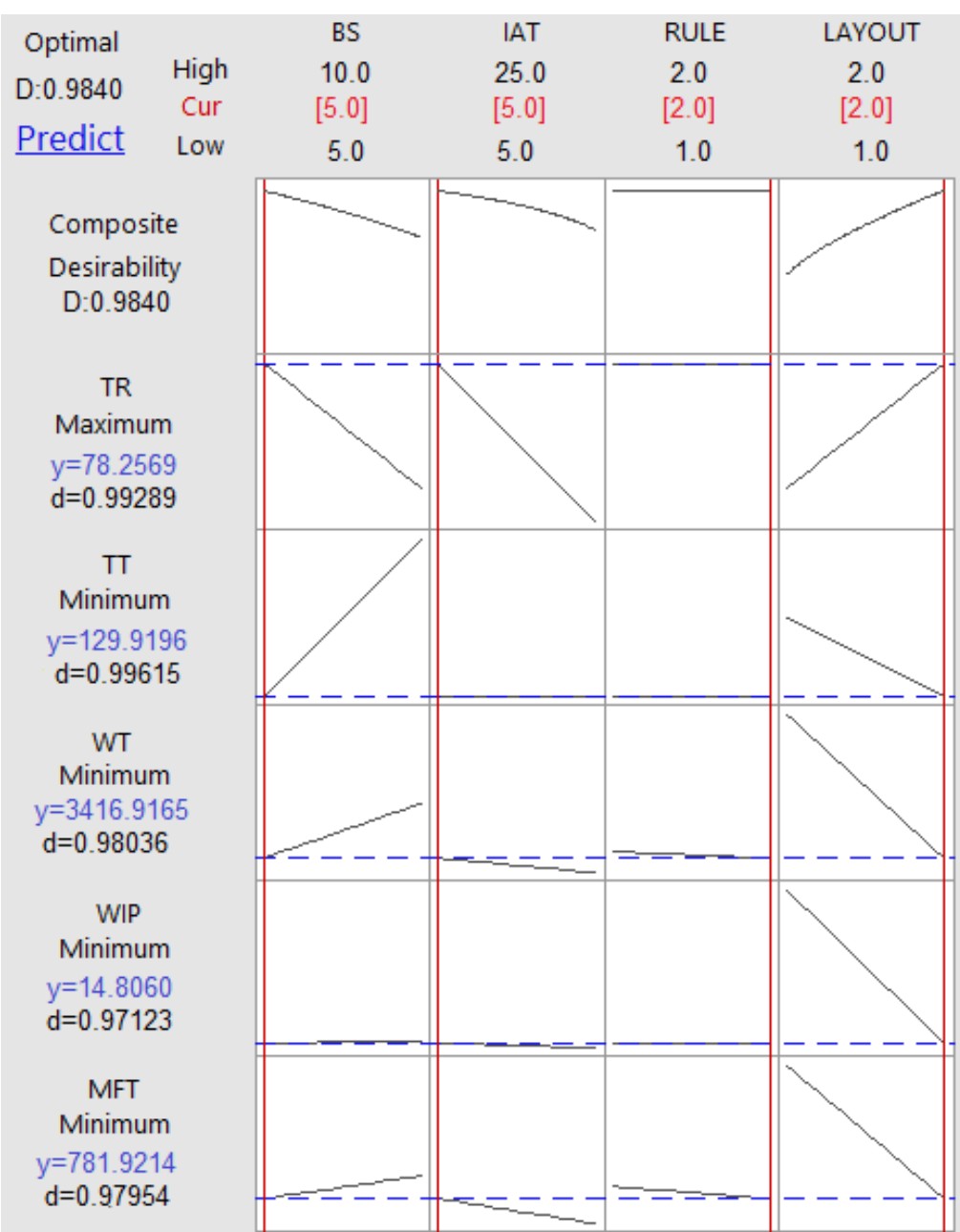

**Figure 7.** Desirability optimization results.

### 3.2.4. The VIKOR Method

Based on Equations (18)–(26), the utility and the regret measures as well as the VIKOR index were computed (Table 8). Once the VIKOR index was ranked, it appears that the optimum performance measures were obtained for factor levels LAYOUT = CL, RULE = SPT, IAT = 25, and BS = 5. The row in bold in Table 8 indicates the optimal solution obtained using VIKOR method, which has a rank equal to 1.

**Table 7.** GRA results.

| Exp | Normalization | | | | | GRC | | | | | GRG | Rank |
|---|---|---|---|---|---|---|---|---|---|---|---|---|
| | **MFT** | **WIP** | **TR** | **TT** | **WT** | **MFT** | **WIP** | **TR** | **TT** | **WT** | | |
| 1 | 0.000 | 0.132 | 0.368 | 0.750 | 0.000 | 0.333 | 0.365 | 0.442 | 0.666 | 0.333 | 0.428 | 16 |
| 2 | 0.860 | 0.874 | 0.354 | 0.003 | 0.715 | 0.781 | 0.799 | 0.436 | 0.334 | 0.637 | 0.597 | 14 |
| 3 | 0.991 | 0.996 | 0.222 | 0.750 | 0.997 | 0.983 | 0.991 | 0.391 | 0.666 | 0.995 | 0.805 | 5 |
| 4 | 0.992 | 0.996 | 0.001 | 0.004 | 0.972 | 0.985 | 0.992 | 0.334 | 0.334 | 0.947 | 0.718 | 11 |
| 5 | 0.864 | 0.000 | 0.390 | 0.744 | 0.773 | 0.786 | 0.333 | 0.451 | 0.661 | 0.688 | 0.584 | 15 |
| 6 | 0.931 | 0.897 | 0.364 | 0.000 | 0.848 | 0.878 | 0.829 | 0.440 | 0.333 | 0.767 | 0.650 | 13 |
| 7 | 0.991 | 0.994 | 0.222 | 0.748 | 0.997 | 0.983 | 0.989 | 0.391 | 0.665 | 0.995 | 0.805 | 6 |
| 8 | 0.992 | 0.996 | 0.002 | 0.001 | 0.972 | 0.985 | 0.992 | 0.334 | 0.334 | 0.947 | 0.718 | 12 |
| 9 | 0.968 | 0.965 | 0.996 | 0.999 | 0.970 | 0.940 | 0.934 | 0.993 | 0.997 | 0.943 | 0.961 | 2 |
| 10 | 0.958 | 0.956 | 0.385 | 0.498 | 0.899 | 0.922 | 0.919 | 0.448 | 0.499 | 0.832 | 0.724 | 10 |
| 11 | 1.000 | 1.000 | 0.224 | 1.000 | 1.000 | 1.000 | 1.000 | 0.392 | 0.999 | 1.000 | 0.878 | 4 |
| 12 | 0.979 | 0.989 | 0.000 | 0.499 | 0.940 | 0.959 | 0.979 | 0.333 | 0.499 | 0.892 | 0.733 | 8 |
| **13** | **0.978** | **0.968** | **1.000** | **0.999** | **0.979** | **0.959** | **0.940** | **1.000** | **0.999** | **0.960** | **0.972** | **1** |
| 14 | 0.959 | 0.955 | 0.386 | 0.496 | 0.901 | 0.924 | 0.917 | 0.449 | 0.498 | 0.835 | 0.725 | 9 |
| 15 | 1.000 | 1.000 | 0.224 | 1.000 | 1.000 | 1.000 | 0.999 | 0.392 | 1.000 | 1.000 | 0.878 | 3 |
| 16 | 0.979 | 0.989 | 0.000 | 0.499 | 0.940 | 0.959 | 0.979 | 0.333 | 0.499 | 0.893 | 0.733 | 7 |

**Table 8.** VIKOR results.

| Exp. | The Utility Measure ($S_i$) | The Regret Measure ($R_i$) | The VIKOR Index ($Q_i$) | Rank |
|---|---|---|---|---|
| 1 | 0.305 | 0.150 | 0.000 | 16 |
| 2 | 0.646 | 0.175 | 0.493 | 14 |
| 3 | 0.947 | 0.200 | 0.959 | 3 |
| 4 | 0.715 | 0.199 | 0.787 | 7 |
| 5 | 0.603 | 0.173 | 0.443 | 15 |
| 6 | 0.691 | 0.186 | 0.639 | 13 |
| 7 | 0.946 | 0.200 | 0.958 | 4 |
| 8 | 0.593 | 0.199 | 0.699 | 12 |
| 9 | 0.903 | 0.200 | 0.928 | 6 |
| 10 | 0.818 | 0.192 | 0.785 | 9 |
| 11 | 1.000 | 0.200 | 1.000 | 2 |
| 12 | 0.681 | 0.198 | 0.749 | 10 |
| 13 | 0.908 | 0.200 | 0.932 | 5 |
| 14 | 0.817 | 0.192 | 0.786 | 8 |
| **15** | **1.000** | **0.200** | **1.000** | **1** |
| 16 | 0.681 | 0.198 | 0.749 | 11 |

## 4. Discussion

The application of the four optimization methods in the context of FMS shows good results for four of the five performances in the case of the two methods GP and VIKOR, and only two performance measures for DF and GRA methods (Table 9).

The results show that the MFT, WIP, TT, and WT performance measures met their targets for GP and VIKOR methods. Indeed, they all show relatively minor deviations from their target values. Only the deviation of RT reaches, respectively, −63.53% and −81.87% in the case of these two methods. On the other hand, in the case of DF and GRA methods, only the optimal values of TT and TR were close to their corresponding targets, while the deviations between the achieved values and the objective values in the case of WIP, WT, and MFT can reach +242.529%, +107.377%, and +83.293% respectively. Hence, the optimization results can be considered satisfactory in the case of GP and VIKOR methods. Meanwhile, it was not the case for the optimization results of the DF and GRA methods.

**Table 9.** MOSO results.

| Performance Measure | | MFT | WIP | TR | TT | WT |
|---|---|---|---|---|---|---|
| Goal value | | 426.596 | 4.323 | 78.303 | 129.757 | 1647.683 |
| GP | Optimized value | 433.000 | 4.425 | 28.557 | 129.922 | 1704.500 |
| | Deviation value | +6.404 | +0.102 | −49.746 | +0.165 | +56.817 |
| | Deviation in % | +1.501% | +2.359% | −63.530% | +0.127% | +3.448% |
| DF | Optimized value | 781.921 | 14.806 | 78.257 | 129.971 | 3416.917 |
| | Deviation value | +355.325 | +10.483 | −0.046 | +0.214 | +1769.234 |
| | Deviation in % | +83.293% | +242.494% | −0.059% | +0.165% | +107.377% |
| GRA | Optimized value | 781.921 | 14.806 | 78.303 | 129.917 | 3416.917 |
| | Deviation value | +355.325 | +10.484 | 0 | +0.160 | +1769.234 |
| | Deviation in % | +83.293% | +242.529% | 0% | +0.124% | +107.377% |
| VIKOR | Optimized value | 428.164 | 4.409 | 14.195 | 129.757 | 1655.257 |
| | Deviation value | +1.568 | +0.086 | −64.108 | 0 | +7.574 |
| | Deviation in % | +0.368% | +1.982% | −81.872% | 0% | +0.460% |

The two methods GP and DF require a higher level of analysis effort than the two methods of GRA and VIKOR. Indeed, in addition to the modeling and development of the simulation models, which is a common point to the four compared optimization methods, as well as the planning of experiments with the DoE method, the two methods GP and DF require relatively higher levels of expertise in the use of the analysis software MINITAB and LINGO. On the opposite side, the two methods GRA and VIKOR only need the development of the equations on Excel, which is within the reach of the majority of DMs. This has an impact on the applicability of the MOSO method.

Table 10 summarizes the performance of the four MSOSO methods being compared in this study. Signs '+' and "−" are assigned to the optimization methods based on their achieved optimization results and their applicability. A '+' is assigned to each method resulting in a good optimization result, which is expressed by reasonable or small deviations. On the other hand, a "−" is assigned to each method that leads to an optimization result characterized by high deviations. For applicability, a "−" is assigned to each method that requires a high level of analysis and expertise. In the opposite case, a '+' is assigned to this optimization method. These methods are then classified according to assigned signs. Any method obtaining two "+" signs will be considered the most efficient. On the other hand, if it obtains two "−" signs, it will be considered as the most mediocre one. In the case where the optimization method obtains both signs "+" and "−", the classification gives priority to the obtained optimization result. The best method is VIKOR, which belongs to group B in the proposed classification. It is followed by the GP method, from group A, since it reaches good optimization results, although it requires a considerable analysis effort. The GRA method, from group B, comes in third rank and the DF method, from the group A, closes the classification at the last rank. This classification shows that the use of optimization methods based on a metamodel does not always produce the best results.

**Table 10.** MOSO performances.

| MOSO | Group | Optimization Result | Applicability | Rank |
|---|---|---|---|---|
| GP | A | + | − | 2 |
| DF | A | − | − | 4 |
| GRA | B | − | + | 3 |
| VIKOR | B | + | + | 1 |

## 5. Conclusions

Various MOSO methods have been presented, developed, and used in the literature. These methods have been the subject of numerous classifications. However, the performance of these methods is not guaranteed due to the lack of comparative studies. Moreover, these classifications have been very diverse and are rarely related to the specific domain of manufacturing systems.

The objective of this research is two-fold. First, we proposed a new conceptual classification of MOSO methods applied to the context of MFS design. Second, four MOSO methods are selected according to this classification and compared through a case study related to the FMS design problem inspired by the literature. This comparison is based on the quality of the optimal solutions obtained by these methods as well as the degree of difficulty of their applicability through the necessary analysis effort and the degree of expertise of the user of these methods. All these studied methods are based on DoE. Two of them are metamodel-based approaches that incorporate the GP and the DF, respectively. The other two methods are not metamodel-based approaches and incorporate GRA and VIKOR, respectively. The comparative results show that the VIKOR method can result in a better optimization than GP, GRA, and DF methods in that order. It is clear, thus, that the use of MOSOs based on meta-models does not produce the best solution in all situations.

This research compares four MOSO methods applied in the context of FMS design. Some future research perspectives should be addressed:

- In this study, four MOSO methods are compared. Two methods belong to group A of the proposed new classification, while the other two belong to group B. The extension of the current comparison to other MOSO methods belonging to group C is the first objective of our interesting perspectives.
- The studied MOSO methods have been applied on a model of an FMS inspired from the literature. This model has six machines grouped in two cells in the CL and three departments in FL. In addition, this FMS processes only four products grouped into two families. Extending the comparison performed in this study to real and more complex FMSs to evaluate the reliability of MOSO methods is the second objective of our interesting perspectives.
- The experimental design developed in this comparison study and which is the basis for the simulation results used in the analysis and generation of optimization solutions is based on four factors: IAT, BS, RULE, and LAYOUT. These four factors are explored on the basis of two levels each. This number of factors and levels remains relatively limited and generates a limited number of experiments. The comparison of MOSO Methods in Manufacturing Systems characterized by a large number of factors and levels is the third objective of our interesting perspectives.
- The application of the compared MOSO methods proceeds through different steps to generate optimization solutions. These steps usually require the intervention of a user to transfer the results from one step to another. The integration of these analysis and optimization steps into the simulation software, as in the case of the OptQuest tool in several simulation tools, would be a very interesting perspective.

**Author Contributions:** Conceptualization, A.J. and W.H.; methodology, A.J. and W.H.; software, A.J.; validation, A.J., N.K.M., A.M.A. and F.M.; formal analysis, W.H. and A.M.A.; investigation, A.J. and N.K.M.; resources, A.J. and N.K.M.; data curation, A.J., A.M.A. and F.M.; writing—original draft preparation, A.J. and F.M.; writing—review and editing, A.J., A.M.A., N.K.M. and W.H.; visualization, A.J. and W.H.; supervision, W.H. and F.M.; project administration, A.J., A.M.A., W.H. and F.M.; funding acquisition, A.M.A. All authors have read and agreed to the published version of the manuscript.

**Funding:** This research was supported and funded by Taif University Researchers Supporting Project number (TURSP-2020/229), Taif University, Taif, Saudi Arabia.

**Institutional Review Board Statement:** Not applicable.

**Informed Consent Statement:** Not applicable.

**Data Availability Statement:** Data are contained within the article.

**Acknowledgments:** This research was supported by Taif University Researchers Supporting Project number (TURSP-2020/229), Taif University, Taif, Saudi Arabia. First, the authors are grateful for the financial support. Second, the authors would like to thank the editors and the three anonymous referees for their valuable and constructive comments on the first draft of this manuscript.

**Conflicts of Interest:** The authors declare no conflict of interest.

## Appendix A

**Table A1.** MFT simulation results according to DoE.

| EXP | Factor Levels | | | | Replication | | | | | | | | | |
| | LAYOUT | RULE | IAT | BS | 1 | 2 | 3 | 4 | 5 | 6 | 7 | 8 | 9 | 10 |
|---|---|---|---|---|---|---|---|---|---|---|---|---|---|---|
| 1 | 1 | 1 | 5 | 5 | 16,334.1 | 16,645.9 | 16,612.2 | 16,866.2 | 17,446.5 | 16,523.3 | 16,778.2 | 16,601.8 | 17,529.3 | 18,119.7 |
| 2 | 1 | 1 | 5 | 10 | 2312.8 | 1636.8 | 2435.7 | 3132.7 | 2374.0 | 3222.1 | 3128.3 | 1968.7 | 4831.2 | 2432.0 |
| 3 | 1 | 1 | 25 | 5 | 597.0 | 549.5 | 559.6 | 528.5 | 614.4 | 540.6 | 583.7 | 577.4 | 552.6 | 569.8 |
| 4 | 1 | 1 | 25 | 10 | 538.0 | 565.0 | 560.4 | 577.7 | 559.5 | 581.1 | 546.9 | 538.7 | 538.2 | 547.6 |
| 5 | 1 | 2 | 5 | 5 | 2574.7 | 3339.8 | 2014.4 | 2735.9 | 2348.5 | 2484.6 | 2698.9 | 1959.8 | 2784.8 | 3760.3 |
| 6 | 1 | 2 | 5 | 10 | 1881.7 | 1261.2 | 1463.6 | 1614.9 | 1166.6 | 1919.4 | 1276.1 | 1449.3 | 1662.2 | 2019.3 |
| 7 | 1 | 2 | 25 | 5 | 547.0 | 570.4 | 567.4 | 564.8 | 552.6 | 559.2 | 599.0 | 565.1 | 606.4 | 577.2 |
| 8 | 1 | 2 | 25 | 10 | 546.7 | 544.0 | 545.1 | 561.6 | 544.9 | 565.4 | 543.0 | 558.3 | 559.2 | 562.0 |
| 9 | 2 | 1 | 5 | 5 | 1460.9 | 1149.0 | 1314.0 | 711.9 | 825.9 | 699.7 | 848.5 | 846.1 | 832.2 | 859.4 |
| 10 | 2 | 1 | 5 | 10 | 1087.7 | 1052.0 | 1305.7 | 993.0 | 1214.9 | 1155.9 | 1038.4 | 1132.8 | 1153.9 | 1085.4 |
| 11 | 2 | 1 | 25 | 5 | 434.5 | 431.8 | 423.1 | 422.7 | 419.7 | 427.9 | 423.9 | 423.9 | 424.8 | 433.5 |
| 12 | 2 | 1 | 25 | 10 | 773.0 | 769.2 | 796.1 | 783.9 | 771.2 | 768.1 | 772.8 | 769.6 | 791.4 | 783.7 |
| 13 | 2 | 2 | 5 | 5 | 690.2 | 717.1 | 752.2 | 721.7 | 1260.2 | 741.1 | 721.4 | 748.1 | 690.2 | 777.0 |
| 14 | 2 | 2 | 5 | 10 | 1159.5 | 1053.5 | 1133.9 | 1121.7 | 1080.1 | 1187.4 | 1071.3 | 1085.1 | 1080.9 | 1094.4 |
| 15 | 2 | 2 | 25 | 5 | 438.0 | 427.5 | 421.1 | 422.6 | 428.0 | 425.6 | 423.8 | 426.3 | 436.3 | 432.3 |
| 16 | 2 | 2 | 25 | 10 | 773.0 | 769.0 | 796.1 | 783.9 | 771.0 | 768.1 | 772.8 | 769.6 | 791.4 | 783.4 |

LAYOUT: 1 = FL, 2 = C; RULE: 1 = FCFS, 2 = SPT.

**Table A2.** WIP simulation results according to DoE.

| EXP | Factor Levels | | | | Replication | | | | | | | | | |
| | LAYOUT | RULE | IAT | BS | 1 | 2 | 3 | 4 | 5 | 6 | 7 | 8 | 9 | 10 |
|---|---|---|---|---|---|---|---|---|---|---|---|---|---|---|
| 1 | 1 | 1 | 5 | 5 | 276.4 | 285.9 | 287.2 | 288.6 | 296.7 | 286.0 | 286.4 | 288.0 | 299.9 | 308.5 |
| 2 | 1 | 1 | 5 | 10 | 38.5 | 27.3 | 41.1 | 52.3 | 39.8 | 53.7 | 52.1 | 33.1 | 80.0 | 40.6 |
| 3 | 1 | 1 | 25 | 5 | 6.1 | 5.6 | 5.7 | 5.4 | 6.2 | 5.5 | 5.9 | 5.9 | 5.6 | 5.8 |
| 4 | 1 | 1 | 25 | 10 | 5.4 | 5.7 | 5.6 | 5.8 | 5.6 | 5.9 | 5.5 | 5.4 | 5.4 | 5.5 |
| 5 | 1 | 2 | 5 | 5 | 325.0 | 345.3 | 302.0 | 349.1 | 318.0 | 322.5 | 332.9 | 329.0 | 371.0 | 343.7 |
| 6 | 1 | 2 | 5 | 10 | 58.2 | 23.5 | 38.3 | 29.4 | 22.4 | 65.8 | 22.7 | 35.2 | 48.4 | 38.5 |
| 7 | 1 | 2 | 25 | 5 | 5.9 | 6.2 | 6.2 | 6.1 | 6.0 | 6.0 | 6.5 | 6.1 | 6.6 | 6.3 |
| 8 | 1 | 2 | 25 | 10 | 5.6 | 5.6 | 5.6 | 5.7 | 5.6 | 5.8 | 5.5 | 5.7 | 5.7 | 5.7 |
| 9 | 2 | 1 | 5 | 5 | 24.4 | 19.2 | 22.0 | 12.0 | 13.9 | 11.7 | 14.2 | 14.2 | 14.0 | 14.4 |
| 10 | 2 | 1 | 5 | 10 | 18.2 | 17.6 | 21.9 | 16.6 | 20.3 | 19.3 | 17.4 | 18.9 | 19.3 | 18.1 |
| 11 | 2 | 1 | 25 | 5 | 4.4 | 4.4 | 4.3 | 4.3 | 4.3 | 4.3 | 4.3 | 4.3 | 4.3 | 4.4 |
| 12 | 2 | 1 | 25 | 10 | 7.8 | 7.7 | 8.0 | 7.9 | 7.7 | 7.7 | 7.8 | 7.7 | 8.0 | 7.9 |
| 13 | 2 | 2 | 5 | 5 | 14.4 | 13.1 | 14.2 | 13.2 | 24.0 | 13.6 | 16.3 | 13.7 | 12.5 | 13.0 |
| 14 | 2 | 2 | 5 | 10 | 20.1 | 18.1 | 19.7 | 19.4 | 18.6 | 20.6 | 19.1 | 18.7 | 18.7 | 18.8 |
| 15 | 2 | 2 | 25 | 5 | 4.5 | 4.4 | 4.3 | 4.3 | 4.4 | 4.4 | 4.4 | 4.4 | 4.5 | 4.5 |
| 16 | 2 | 2 | 25 | 10 | 7.8 | 7.8 | 8.0 | 7.9 | 7.8 | 7.8 | 7.8 | 7.8 | 8.0 | 7.9 |

LAYOUT: 1 = FL, 2 = C; RULE: 1 = FCFS, 2 = SPT.

**Table A3.** TR simulation results according to DoE.

| EXP | Factor Levels | | | | Replication | | | | | | | | | |
|-----|--------|------|-----|-----|------|------|------|------|------|------|------|------|------|------|
|     | LAYOUT | RULE | IAT | BS  | 1    | 2    | 3    | 4    | 5    | 6    | 7    | 8    | 9    | 10   |
| 1   | 1 | 1 | 5  | 5  | 39.54 | 38.37 | 37.72 | 38.23 | 37.29 | 37.99 | 38.23 | 37.76 | 36.74 | 35.8  |
| 2   | 1 | 1 | 5  | 10 | 37.84 | 38.59 | 36.66 | 36.33 | 38.07 | 35.98 | 35.96 | 37.22 | 34.69 | 37.81 |
| 3   | 1 | 1 | 25 | 5  | 28.52 | 28.46 | 28.31 | 28.48 | 28.4  | 28.36 | 28.42 | 28.37 | 28.46 | 28.45 |
| 4   | 1 | 1 | 25 | 10 | 14.31 | 14.31 | 14.31 | 14.26 | 14.31 | 14.26 | 14.31 | 14.28 | 14.24 | 14.27 |
| 5   | 1 | 2 | 5  | 5  | 39.7  | 39.05 | 41.83 | 37.92 | 40.49 | 40.27 | 39.34 | 39.88 | 35.48 | 38.14 |
| 6   | 1 | 2 | 5  | 10 | 36.3  | 38.67 | 37.49 | 38.27 | 38.46 | 35.59 | 39.18 | 37.21 | 36.57 | 37.71 |
| 7   | 1 | 2 | 25 | 5  | 28.52 | 28.48 | 28.35 | 28.36 | 28.41 | 28.45 | 28.37 | 28.41 | 28.33 | 28.28 |
| 8   | 1 | 2 | 25 | 10 | 14.27 | 14.31 | 14.27 | 14.3  | 14.27 | 14.34 | 14.31 | 14.35 | 14.27 | 14.27 |
| 9   | 2 | 1 | 5  | 5  | 76.95 | 77.61 | 77.27 | 78.35 | 78.27 | 78.71 | 78.5  | 78.53 | 78.23 | 78.24 |
| 10  | 2 | 1 | 5  | 10 | 38.79 | 38.88 | 38.56 | 39.05 | 38.91 | 38.88 | 38.89 | 38.91 | 38.87 | 39.12 |
| 11  | 2 | 1 | 25 | 5  | 28.54 | 28.5  | 28.58 | 28.54 | 28.58 | 28.5  | 28.62 | 28.54 | 28.58 | 28.62 |
| 12  | 2 | 1 | 25 | 10 | 14.19 | 14.19 | 14.15 | 14.23 | 14.22 | 14.16 | 14.18 | 14.23 | 14.18 | 14.22 |
| 13  | 2 | 2 | 5  | 5  | 78.5  | 78.52 | 78.24 | 78.47 | 77.5  | 78.6  | 77.39 | 78.63 | 78.72 | 78.46 |
| 14  | 2 | 2 | 5  | 10 | 38.92 | 39.06 | 38.73 | 39.03 | 38.92 | 38.97 | 38.77 | 39.03 | 38.93 | 39.05 |
| 15  | 2 | 2 | 25 | 5  | 28.53 | 28.5  | 28.58 | 28.5  | 28.54 | 28.55 | 28.62 | 28.54 | 28.57 | 28.58 |
| 16  | 2 | 2 | 25 | 10 | 14.19 | 14.19 | 14.15 | 14.23 | 14.22 | 14.16 | 14.18 | 14.23 | 14.18 | 14.22 |

LAYOUT: 1 = FL, 2 = C; RULE: 1 = FCFS, 2 = SPT.

**Table A4.** WT simulation results according to DoE.

| EXP | Factor Levels | | | | Replication | | | | | | | | | |
|-----|--------|------|-----|-----|------|------|------|------|------|------|------|------|------|------|
|     | LAYOUT | RULE | IAT | BS  | 1    | 2    | 3    | 4    | 5    | 6    | 7    | 8    | 9    | 10   |
| 1   | 1 | 1 | 5  | 5  | 82,836.8 | 85,752.2 | 86,140.3 | 86,578.0 | 89,250.2 | 84,996.3 | 85,347.6 | 85,214.9 | 89,138.4 | 92,596.2 |
| 2   | 1 | 1 | 5  | 10 | 21,801.7 | 15,089.4 | 22,715.8 | 29,844.8 | 22,789.5 | 30,417.2 | 29,423.0 | 18,018.8 | 46,237.5 | 23,032.6 |
| 3   | 1 | 1 | 25 | 5  | 1927.4 | 1838.2 | 1837.4 | 1681.1 | 2073.2 | 1759.2 | 1939.3 | 1923.3 | 1779.0 | 1920.5 |
| 4   | 1 | 1 | 25 | 10 | 3938.8 | 4089.8 | 4115.0 | 4115.8 | 4030.9 | 4087.6 | 3949.3 | 3991.3 | 4038.8 | 3955.6 |
| 5   | 1 | 2 | 5  | 5  | 26,779.8 | 25,064.3 | 14,124.1 | 24,226.4 | 13,939.3 | 19,553.4 | 21,059.2 | 12,639.7 | 14,090.3 | 37,979.1 |
| 6   | 1 | 2 | 5  | 10 | 17,439.0 | 11,361.6 | 13,119.9 | 15,247.0 | 10,391.8 | 19,065.1 | 11,499.0 | 13,196.1 | 15,287.0 | 19,011.7 |
| 7   | 1 | 2 | 25 | 5  | 1776.7 | 1854.7 | 1843.8 | 1829.6 | 1754.0 | 1829.4 | 1942.5 | 1837.2 | 2073.1 | 1883.9 |
| 8   | 1 | 2 | 25 | 10 | 4019.1 | 4010.4 | 4017.5 | 4086.7 | 4034.9 | 4129.1 | 3973.3 | 4076.8 | 4030.6 | 4063.8 |
| 9   | 2 | 1 | 5  | 5  | 6628.1 | 5230.5 | 5937.7 | 3069.7 | 3627.5 | 3008.1 | 3746.4 | 3740.8 | 3663.6 | 3781.3 |
| 10  | 2 | 1 | 5  | 10 | 10,025.4 | 9574.9 | 12,052.1 | 8981.6 | 11,220.2 | 10,425.2 | 9458.4 | 10,387.8 | 10,438.4 | 9835.0 |
| 11  | 2 | 1 | 25 | 5  | 1664.4 | 1655.0 | 1637.7 | 1637.6 | 1634.2 | 1676.7 | 1646.6 | 1630.7 | 1640.7 | 1653.1 |
| 12  | 2 | 1 | 25 | 10 | 6773.5 | 6798.3 | 6731.1 | 6853.6 | 6657.2 | 6739.5 | 6990.0 | 6786.5 | 6747.4 | 6830.7 |
| 13  | 2 | 2 | 5  | 5  | 3367.2 | 3093.6 | 3233.1 | 3102.4 | 5799.1 | 3205.1 | 3084.3 | 3216.5 | 3006.0 | 3062.0 |
| 14  | 2 | 2 | 5  | 10 | 10,596.6 | 9474.9 | 10,320.7 | 10,257.9 | 9831.7 | 10,804.9 | 9753.2 | 9931.9 | 9828.1 | 9990.0 |
| 15  | 2 | 2 | 25 | 5  | 1671.3 | 1631.5 | 1636.3 | 1632.5 | 1646.6 | 1654.3 | 1646.3 | 1651.4 | 1693.2 | 1689.2 |
| 16  | 2 | 2 | 25 | 10 | 6772.9 | 6796.7 | 6730.7 | 6853.6 | 6654.9 | 6739.5 | 6707.0 | 6786.5 | 6747.4 | 6827.4 |

LAYOUT: 1 = FL, 2 = C; RULE: 1 = FCFS, 2 = SPT.

**Table A5.** TT simulation results according to DoE.

| EXP | Factor Levels | | | | Replication | | | | | | | | | |
|-----|--------|------|-----|-----|------|------|------|------|------|------|------|------|------|------|
|     | LAYOUT | RULE | IAT | BS  | 1    | 2    | 3    | 4    | 5    | 6    | 7    | 8    | 9    | 10   |
| 1   | 1 | 1 | 5  | 5  | 194.0 | 194.6 | 195.2 | 195.2 | 196.3 | 194.5 | 195.0 | 195.2 | 195.6 | 194.2 |
| 2   | 1 | 1 | 5  | 10 | 389.9 | 389.1 | 390.2 | 389.2 | 387.4 | 389.8 | 389.4 | 389.2 | 390.6 | 388.4 |
| 3   | 1 | 1 | 25 | 5  | 194.7 | 195.4 | 195.0 | 194.7 | 195.7 | 195.5 | 195.4 | 194.9 | 194.7 | 194.1 |
| 4   | 1 | 1 | 25 | 10 | 388.3 | 389.2 | 388.9 | 390.1 | 389.0 | 389.7 | 390.0 | 389.6 | 386.9 | 388.8 |
| 5   | 1 | 2 | 5  | 5  | 195.7 | 195.2 | 194.6 | 197.1 | 193.3 | 193.9 | 195.2 | 210.8 | 194.0 | 194.4 |
| 6   | 1 | 2 | 5  | 10 | 392.3 | 387.7 | 390.0 | 388.1 | 390.3 | 392.1 | 389.9 | 390.8 | 390.7 | 389.5 |

| EXP | Factor Levels | | | | Replication | | | | | | | | | |
|-----|--------|------|-----|----|-------|-------|-------|-------|-------|-------|-------|-------|-------|-------|
| | LAYOUT | RULE | IAT | BS | 1 | 2 | 3 | 4 | 5 | 6 | 7 | 8 | 9 | 10 |
| 7 | 1 | 2 | 25 | 5 | 195.5 | 194.8 | 195.1 | 195.9 | 195.2 | 195.4 | 194.9 | 195.3 | 195.3 | 195.5 |
| 8 | 1 | 2 | 25 | 10 | 391.7 | 388.7 | 390.9 | 389.4 | 389.7 | 390.2 | 388.3 | 387.6 | 392.7 | 390.1 |
| 9 | 2 | 1 | 5 | 5 | 130.0 | 130.1 | 130.3 | 130.4 | 129.9 | 129.9 | 129.7 | 130.7 | 129.7 | 130.3 |
| 10 | 2 | 1 | 5 | 10 | 262.1 | 259.5 | 261.0 | 260.6 | 260.6 | 259.9 | 258.8 | 261.0 | 260.5 | 259.6 |
| 11 | 2 | 1 | 25 | 5 | 130.2 | 129.0 | 130.2 | 130.5 | 129.4 | 130.1 | 129.2 | 130.3 | 129.6 | 130.4 |
| 12 | 2 | 1 | 25 | 10 | 258.4 | 258.8 | 261.2 | 259.6 | 261.2 | 260.4 | 261.0 | 259.3 | 262.8 | 260.4 |
| 13 | 2 | 2 | 5 | 5 | 129.4 | 129.7 | 129.8 | 129.9 | 129.8 | 130.4 | 130.1 | 129.9 | 130.3 | 129.9 |
| 14 | 2 | 2 | 5 | 10 | 262.6 | 260.9 | 259.8 | 261.4 | 260.5 | 261.7 | 260.1 | 260.8 | 262.4 | 260.7 |
| 15 | 2 | 2 | 25 | 5 | 130.0 | 128.9 | 130.0 | 130.2 | 130.0 | 129.9 | 129.2 | 129.9 | 129.4 | 130.2 |
| 16 | 2 | 2 | 25 | 10 | 258.4 | 258.8 | 261.2 | 259.6 | 261.2 | 260.4 | 261.0 | 259.3 | 262.8 | 260.4 |

LAYOUT: 1 = FL, 2 = C; RULE: 1 = FCFS, 2 = SPT.

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
