# Peer review of "Multi-Objective Design Optimization of Flexible Manufacturing Systems Using Design of Simulation Experiments: A Comparative Study"

_machines, doi:10.3390/machines10040247_

Round 1

Reviewer 1 Report

I have these comments only:

1- Please discuss in the text different classifications of multi-objective algorithms: a priori, a posterior, and interactive 

2- Where is the Pareto optimal front and solutions obtained? 

3- Any way to provide convergence curves?

Author Response

Responses to the reviewer 1 comment's

Manuscript ID: machines-1638467

Type: Article

Title: Multi-Objective Design Optimization of Flexible Manufacturing Systems Using Designing Simulation Experiments: A Comparative Study

Authors:  Abdessalem Jerbi, Wafik Hachicha, Awad M. Aljuaid, Neila Khabou Masmoudi Faouzi Masmoudi

The authors would like to thank the editor and the anonymous reviewers, whose insightful comments and constructive suggestions helped us to significantly improve the quality of this paper. Every change in the text is colored in red.

Reviewer 2 Report

- It is never stated what are the metamodels that are sued, several of these exist in literature with significantly different performance. Therefore, it is important to make clear what metamodel is iused.

- In general the nomenclature is poorly treated, with terms that are never made explicated or changes (also between capital and normal font) within the paper.

- It is not clear how to interpret Figure 7.

- Same number of significant figures.

- The results in Table 9 and those discussed between lines 641-655 are not consistent.

- In table 10, how the plus and minus sign are decided?

- There are no results about the behavior/convergence of the different algorithms. Neither a single Pareto solution set is reported.

Author Response

Responses to the reviewer 2 comment's

Manuscript ID: machines-1638467

Type: Article

Title: Multi-Objective Design Optimization of Flexible Manufacturing Systems Using Designing Simulation Experiments: A Comparative Study

Authors:  Abdessalem Jerbi, Wafik Hachicha, Awad M. Aljuaid, Neila Khabou Masmoudi Faouzi Masmoudi

The authors would like to thank the editor and the anonymous reviewers, whose insightful comments and constructive suggestions helped us to significantly improve the quality of this paper. Every change in the text is colored in red.

Reviewer 3 Report

Clearly processed comparative study of using of designing simulation experiments for multi-objective design optimization of flexible manufacturing systems. The authors identified uncovered areas in the field of comparison different MOSO methods and offer a study of several relatively straightforward simulation-based FM optimization methodologies that cover different categories of optimization methods classification, represented by four selected methods, all based on the DoE technique.

The structure of the article is clear, the goals and starting points are clearly defined, the proposed solution is unambiguous and the achieved results are correctly evaluated.
The strengths of the article are:
- clearly processed current state of the selected area based on its own classification of MOSO for FMS design;
- clearly identified gaps in published outputs;
- comparison of four selected methods.
I also consider the clearly defined future direction of research to be a great benefit, which gives a great precondition for further follow-up creditworthy scientific contributions.

In order to increase the clarity of the achieved results, it would be appropriate to highlight the Rank 1 results in the Table 7 and Table 8.

I have no other reservations about the form or the content of the submitted manuscript. 

Author Response

Responses to the reviewer 3 comment's

Manuscript ID: machines-1638467

Type: Article

Title: Multi-Objective Design Optimization of Flexible Manufacturing Systems Using Designing Simulation Experiments: A Comparative Study

Authors:  Abdessalem Jerbi, Wafik Hachicha, Awad M. Aljuaid, Neila Khabou Masmoudi Faouzi Masmoudi

The authors would like to thank the editor and the anonymous reviewers, whose insightful comments and constructive suggestions helped us to significantly improve the quality of this paper. Every change in the text is colored in red.
